# Reprogramming of myeloid cells and their progenitors in patients with non-medullary thyroid carcinoma

Katrin Rabold[1,2,3,16], Martijn Zoodsma [4,5,16], Inge Grondman[1], Yunus Kuijpers [4,5], Manita Bremmers[6], Martin Jaeger[1,3], Bowen Zhang[4,5], Willemijn Hobo[7], Han J. Bonenkamp[8], Johannes H. W. de Wilt [8], Marcel J. R. Janssen[9], Lenneke A. M. Cornelissen[2], Ilse C. H. van Engen-van Grunsven[10], Willem J. M. Mulder [11,12,13], Jan W. A. Smit[14], Gosse J. Adema [2], Mihai G. Netea [1,3,15], Yang Li [1,3,4,5], Cheng-Jian Xu[1,3,4,5] ✉ & Romana T. Netea-Maier [14] ✉

Myeloid cells, crucial players in antitumoral defense, are affected by tumor-derived factors and treatment. The role of myeloid cells and their progenitors prior to tumor infiltration is poorly understood. Here we show single-cell transcriptomics and functional analyses of the myeloid cell lineage in patients with non-medullary thyroid carcinoma (TC) and multinodular goiter, before and after treatment with radioactive iodine compared to healthy controls. Integrative data analysis indicates that monocytes of TC patients have transcriptional upregulation of antigen presentation, reduced cytokine production capacity, and overproduction of reactive oxygen species. Interestingly, these cancer-related pathological changes are partially removed upon treatment. In bone marrow, TC patients tend to shift from myelopoiesis towards lymphopoiesis, reflected in transcriptional differences. Taken together, distinct transcriptional and functional changes in myeloid cells arise before their infiltration of the tumor and are already initiated in bone marrow, which suggests an active role in forming the tumor immune microenvironment.

Immunotherapy has emerged in the last decade from being an unfulfilled promise to a mainstay cancer treatment next to surgery, chemo- or radiotherapy. Inhibition of immune checkpoints, such as targeting the PD-1/PD-L1 axis or CTLA4 with therapeutic monoclonal antibodies, has been proven effective for sub-groups of patients with several types of cancer[1–3]. These therapies are aimed at improving the response of T lymphocytes against tumor cells, and led to a T-cell-centered view of onco-immunology. However, while T-cell-based immunotherapy has revolutionized treatment of cancer, it is effective only in a minority of patients, and improved approaches are needed.

Accumulating evidence shows that cells of the innate immune system, such as natural killer cells, neutrophils and monocytes, are also crucial components of the immune response to cancer, but with dual roles. On the one hand innate immune cells can directly kill tumor cells and drive the tumor-specific T-cell responses. On the other hand, they can also gain an immunosuppressive phenotype under the influence of the tumor and inhibit the tumor-killing activities of lymphocytes. Here we focus on bone marrow-derived myeloid cells, which can even enhance tumor growth by releasing tumor cell survival factors[4] and enhance tumor supportive inflammation[5]. Accordingly, high infiltration of tumors with tumor-associated macrophages (TAM) is frequently associated with a poor clinical outcome in many tumor types[6].

Research performed in the last years has also shown that different physiological conditions such as mild infections or vaccination[7] or

diseases such as atherosclerosis[8] can induce long-term changes in myeloid cell function, which are already instigated at the level of their progenitors in the bone marrow. These changes subsequently determine the function of the myeloid cells in circulation and organs. While in the case of infections or vaccinations such processes (also called 'trained immunity') improve the function of myeloid cells[9,10], one could hypothesize that opposite effects occur in patients with malignancies in which myeloid cell function is defective. Circumstantial evidence that this may indeed take place is provided by the identification of myeloid-derived suppressor cells in the circulation of cancer patients[11,12], implicating changes at the level of the bone marrow. In line with this, tumors may exert effects on circulating immune cells and the bone marrow by releasing proteins, metabolites, exosomes or even circulating neoplastic cells. Indeed, an earlier study has suggested changes in myelopoiesis in cancer patients[13], but how the transcriptional program and function of monocytes and their progenitors is different in patients with malignancies is largely unknown.

One type of cancer known to be highly infiltrated by TAMs is non-medullary thyroid carcinoma (TC), particularly the poorly differentiated and the undifferentiated (anaplastic) TC, which is associated with an unfavorable clinical outcome[14,15]. These properties of TC make it a suitable model to study myeloid cell behavior in cancer. Conventional treatment of TC is surgical removal of the tumor by total- or hemi-thyroidectomy, in many patients followed by radioactive iodine ($^{131}$I) for remnant ablation. Although effective in most patients, this therapy is much less effective in patients with advanced disease or poorly differentiated tumors. These patients have limited treatment options and their prognosis remains poor[16–18]. It is therefore of paramount importance to identify new treatment strategies for these TC patients. Exploring the role of the innate immune system in TC, especially the role and possible modulation of TAMs, may be rewarding in this respect.

In the present study we integrate single-cell transcriptional and functional analysis of circulating myeloid cells and their bone marrow progenitor cells to characterize the spectrum of immune cell states in patients with TC. As comparators we use both healthy volunteers and patients with multinodular goiter (MNG), representing benign tumors of the thyroid gland that undergo treatments (surgery or radioactive iodine) similar to those applied in patients with malignant TC. Additionally, we assess the effects of surgery and $^{131}$I treatment on myeloid cell function and their tumor suppressive phenotype. We show that in TC, myeloid cells have upregulated antigen presentation-related genes, reduced cytokine production capacity and they over-produce reactive oxygen species, with transcriptional changes already present in extra-tumoral myeloid cells. Our findings emphasize the potential of innate immune cells in development of novel therapeutic targets.

## Results

### Immune dysregulation in peripheral blood of TC patients
Peripheral blood was collected from 14 patients with TC, 14 patients with MNG, and 8 healthy volunteers (Table 1, Fig. 1A). The median age, sex distribution and median BMI was comparable between the groups. Of all 14 TC patients included, ten were diagnosed with papillary TC (of which one with a microcarcinoma) and four patients were diagnosed with follicular TC (of which three with Hürthle cell carcinoma). Two of the TC patients deceased within three years after diagnosis (Table 1). For some of the immunological assays performed, not enough biological material was available for some of the patients, and subsequently the final number of tests may differ for the various assessments. The precise sample size is disclosed for each type of assay. No sample or outlier removal was performed for any of the immunological functional assays.

Complete blood counts were measured in EDTA blood samples. Whole blood leukocyte (WBC), neutrophil, lymphocyte and monocyte counts did not differ significantly between the groups at baseline

(Fig. 1B). In the single-cell RNA sequencing (scRNA-seq) analysis of PBMCs, we analyzed a total of 16,161 cells from 24 samples (TC $n = 10$, MNG $n = 9$, HC $n = 5$), with an average of 673 cells per sample. Unsupervised clustering identified 14 cell type clusters which were annotated by using known marker gene expression (Fig. 1C, Supplementary Fig. 1A). Comparing the cell proportions of 14 different cell types between the groups revealed no statistically significant differences (Supplementary Fig. 1B).

### Immune cells are transcriptionally different between groups
We further performed differential gene expression analysis, followed by GO term enrichment in cell types that presented sufficient differentially expressed genes between conditions: CD14$^+$ monocytes, CD8$^+$ T cells and naïve CD4$^+$ T cells. For each of these cell types, TC and MNG were separately compared to HC. In the CD14$^+$ monocytes, antigen processing and presentation pathways are upregulated in TC and MNG samples, compared to HC (Fig. 1D), involving several human leukocyte antigen (HLA) genes (Supplementary Fig. 1C). CD8$^+$ T cells show upregulation of T-cell activation and *MHC* expression in the TC group specifically. In the CD4$^+$ T-cell population, TC vs. HC are comparable, whereas the MNG patients show a unique upregulation of several metabolic processes (Fig. 1D).

As these analyses of immune cell populations have shown important changes in the transcriptional profile of myeloid cells in the circulation of TC patients, we next assessed the impact of the malignant process on the immune cell precursors in the bone marrow from which they arise.

### The impact of TC on bone marrow immune cells
Bone marrow was available from 7 patients with TC, 3 patients with MNG, and 5 healthy controls (HC) (Fig. 1A, Table 2). The three groups were comparable in sex distribution and median BMI, but healthy controls were younger than TC patients ($p < 0.05$, Table 2). A total of

**Table 1 | Baseline characteristics blood samples**

| Blood samples | Thyroid carcinoma (n = 14) | Multinodular goiter (n = 14) | Healthy control (n = 8) | p value (all groups) |
|---|---|---|---|---|
| **Age** (years) | | | | 0.0973 |
| Median (IQR) | 60 (52–71) | 70 (55–74) | 53 (48–57) | |
| **Sex** | | | | 0.3241 |
| Male % (n) | 43 (6) | 21 (3) | 50 (4) | |
| Female % (n) | 57 (8) | 79 (11) | 50 (4) | |
| **Body Mass Index** (kg/m²) | | | | 0.2916 |
| Median (IQR) | 26 (24–28) | 28 (24–31) | N/A | |
| **Diagnosis** | | | | N/A |
| Papillary % (n) | 71 (10) | N/A | N/A | |
| Micro-carcinoma % (n) | 7 (1) | N/A | N/A | |
| Follicular % (n) | 29 (4) | N/A | N/A | |
| Hürthle % (n) | 21 (3) | N/A | N/A | |
| **Clinical response** | | | | |
| Deceased within 3 years after diagnosis % (n) | 14 (2) | N/A | N/A | |
| **Dose $^{131}$I** (MBq) | (n = 9) | (n = 8) | | ** |
| Median (IQR) | 7400 (3700–7400) | 1500 (848–2446) | N/A | |

Kruskal–Wallis test (two-sided) followed by Dunn's Multiple-Comparison test or Mann–Whitney U test. Dose $^{131}$I.

*IQR* interquartile range, *MBq* mega Becquerel, *N/A* not applicable.

**$p < 0.01$.

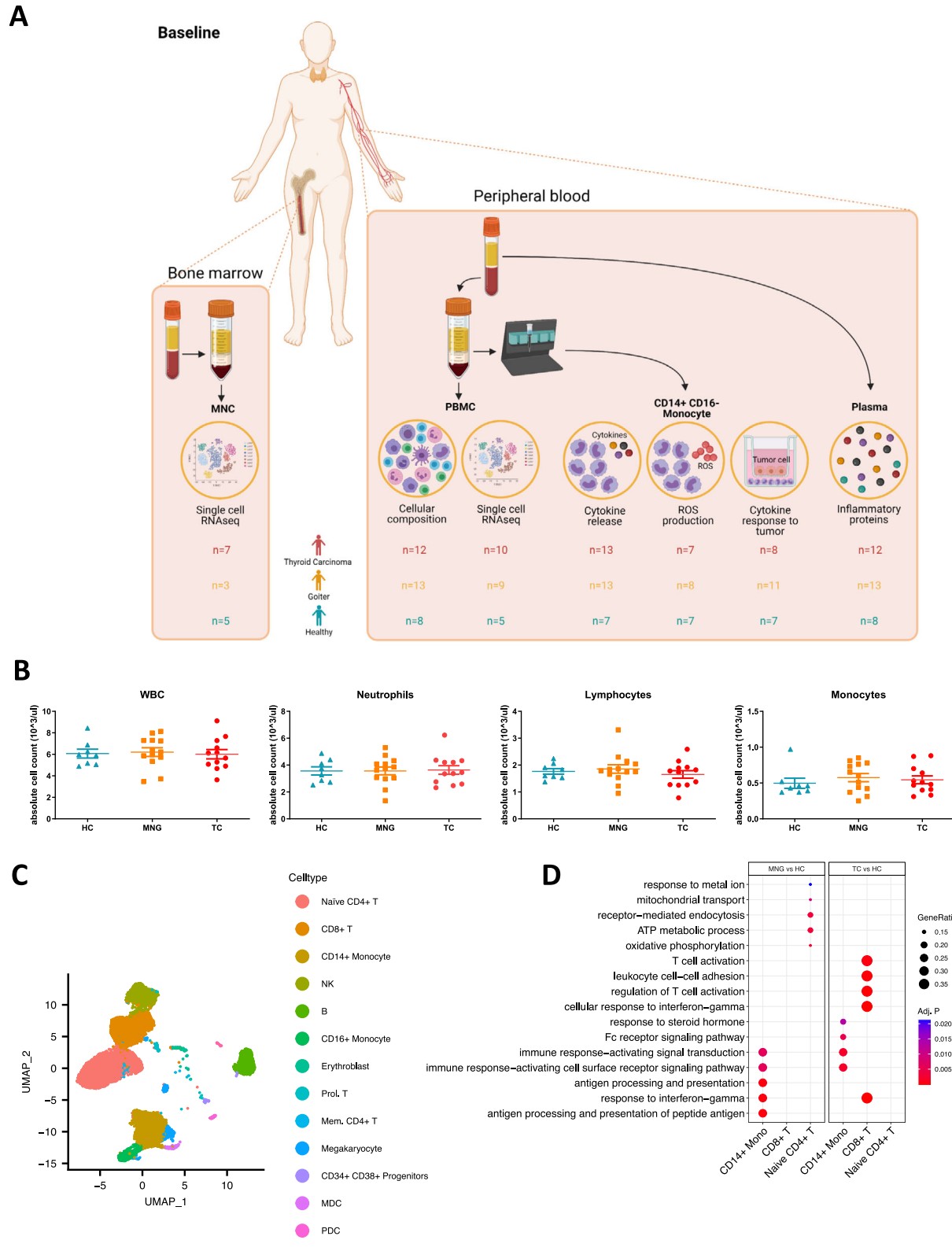

**Fig. 1 | Graphical outline of study design and cell counts and single-cell transcriptomics of peripheral blood mononuclear cells. A** Study Design Baseline Measurements: Bone marrow and Peripheral blood were collected from Thyroid carcinoma (TC) patients, Multinodular Goiter (MNG) patients and Healthy Controls (HC). Illustration created with BioRender.com. **B** Box plot of cell counts in whole blood. (HC $n = 8$, MNG $n = 13$, TC $n = 12$) Mean ± SEM. One-way ANOVA followed by Bonferroni post hoc correction. **C** UMAP projection of the 16,161 PBMC-derived single cells from 24 samples (HC $n = 5$, MNG $n = 9$, TC $n = 10$). All the major cell types were identified using canonical markers (Fig. S1A). **D** GO term enrichment analysis of differentially expressed genes in MNG and TC, compared to HC. Over-representation analysis on the significantly differentially expressed genes for each relevant comparison was used to determine significance. $P$ values were measured using the hypergeometric distribution and adjusted using Benjamini-Hochberg correction. Source data are provided as a Source Data file.

**Table 2 | Baseline characteristics bone marrow samples**

| Bone marrow samples | Thyroid carcinoma (n = 7) | Multinodular goiter (n = 3) | Healthy control (n = 5) | p value (all groups) |
|---|---|---|---|---|
| **Age** (years) | | | | *0.0292* |
| Median (IQR) | 71 (54–78) | 50 (45–77) | 26 (21–48) | *TC vs. HC |
| **Sex** | | | | 0.9611 |
| Male % (n) | 43 (3) | 33 (1) | 40 (2) | |
| Female % (n) | 57 (4) | 66 (2) | 60 (3) | |
| **Body Mass Index** (kg/m²) | | | | 0.5115 |
| Average (s.d.) | 23.8 (3.4) | 25.1 (2.2) | 26.2 (3.4) | |
| **Diagnosis** | | | | N/A |
| Papillary % (n) | 57 (4) | N/A | N/A | |
| Microcarcinoma % (n) | 14 (1) | N/A | N/A | |
| Follicular % (n) | 43 (3) | N/A | N/A | |
| Hürthle % (n) | 29 (2) | N/A | N/A | |
| **Clinical response** | | | | |
| Deceased within 3 years after diagnosis % (n) | 29 (2) | N/A | N/A | |

Kruskal–Wallis test (two-sided) followed by Dunn's Multiple-Comparison test.
*IQR* interquartile range, *N/A* not applicable.
*$p < 0.05$.
Italic values indicate statistical signficance *p* value.

8,588 bone marrow-derived mononuclear cells (BM-MNCs) were analyzed, with an average of 572 cells per sample.

Within the BM-MNCs, CD34 expression was used to identify progenitor cells (Supplementary Fig. 1D). Uniform manifold approximation and projection (UMAP)[19], followed by graph-based clustering identified subpopulations within the CD34+ cell population (Fig. 2A). We manually annotated each cluster with distinct cellular identities and identified myeloid/multipotent progenitors, lymphoid progenitors, early B-cell progenitors, and late B-cell progenitors (Fig. 2B). Positive marker genes were identified for each of the cell subpopulations and used for GO term enrichment, showing a clear distinction between the different progenitor cell types and the more differentiated B-cell progenitors (Fig. 2C). Comparison of the cell subpopulations between the three groups reveals no statistically significant difference in the proportions between groups, likely due to limited sample size. However, we observe a trend of lower proportion of the myeloid/multipotent progenitor cells in the TC and MNG samples compared to healthy controls, whereas a shift towards late B-cell progenitors is observed in the TC samples (Fig. 2D). The tendency towards elevated late B-cell progenitor proportions is unique to TC patients (Fig. 2D). These results have been further validated by flow cytometric analysis (Supplementary Fig. 2). All positive marker genes with a log fold change of 2 or higher were plotted per cell type separated by condition, showing no obvious differences between the conditions (Supplementary Fig. 1E).

We aimed to further dissect the differentiation trajectories of bone marrow-derived progenitor cells. Therefore, we performed pseudo-time analysis to calculate a measure of progress of cell differentiation per cell[20] on the CD34+ bone marrow-derived progenitor cells and PBMC-derived B cells. This identified a trajectory from the myeloid/multipotent progenitor cells, through the lymphoid progenitor cells and late B-cell progenitors towards the B cells, splitting off into two separate branches of PBMC-derived B cells (Fig. 2E, F). This confirms that the bone-marrow-derived cells that were identified as B-cell progenitors are indeed connected to the B-cell lineage in peripheral blood. Dimensionality reduction indicated no substantial phenotypic differences between TC, MNG and HC (Fig. 2G).

Next, we examined transcriptional differences between disease status. Myeloid/multipotent progenitor cells showed small transcriptional differences. Two differentially expressed genes have been identified in the myeloid/multipotent progenitor cells between TC patients and HC, five differentially upregulated genes have been identified in TC compared to MNG, and fourteen differentially downregulated genes in the comparison of MNG versus HC (Fig. 2H). Interestingly, we observed downregulation of the oncogene *AREG*, which plays a role in an inflammatory response[21], in myeloid/multipotent progenitor cells from TC and MNG patients compared to HC.

To obtain a broad overview of the conditions in the periphery and bone marrow compartments, we performed PCA-based dimensionality reduction on pseudo-bulk expression data (summed scRNA-seq reads across cells). BM-MNCs from TC and MNG patients are markedly different compared to HC (Fig. 2I, $p < 0.05$). In contrast, PBMCs did not show a clear separation of the three groups. Following, we calculated differential expression of TC patients versus HC per cell type, for each compartment individually and calculated their enrichment across the compartments for the overlapping cell types. We then investigated whether the TC signature genes in BM-MNCs were significantly more enriched in expression in the periphery between TC and HC. Enrichment in TC compared to HC was tested using a two-sided Wilcoxon test per cell type individually. *P* values were adjusted (BH) to account for the multiple testing problem over all cell types together. Figure 2J shows that differentially expressed genes identified between TC and HC from BM-MNCs are significantly more enriched in expression in PBMC-derived B cells and CD14+ monocytes of TC than those of HC (two-sided Wilcoxon test, all $p < 0.0001$). Interestingly, the PBMC signature genes between TC and HC (Fig. 2J) were also more enriched in expression in BM-MNCs CD14+ monocytes of TC compared to HC (two-sided Wilcoxon test, all $p < 0.00001$). These results suggest shared TC signatures between BM-MNCs and PBMC in CD14+ monocyte at the transcriptome level.

**Peripheral blood monocytes from TC and MNG patients show a reduced cytokine releasing capacity and slightly increased ROS production capacity**

Freshly isolated peripheral blood CD14+ CD16− classical monocytes were stimulated with TLR4 ligand lipopolysaccharide (LPS), TLR2 ligand Pam3CSK4 (P3C) or IL-1α to investigate their cytokine production capacity. Stimulation with LPS showed no differences in TNF-α response, whereas IL-1β responses were significantly reduced in TC patients ($p = 0.019$). IL-1Ra and IL-6 cytokine responses showed the same trends of reduced production in the patients (Fig. 3A). Notably, a gradient of decreasing interleukin cytokine responses could be observed from healthy controls to MNG patients to TC patients. Similar patterns of cytokine responses were observed upon stimulation with P3C. However, upon P3C stimulation, a significant increase ($p = 0.026$) in IL-6 was observed in MNG patients compared to TC patients (Supplementary Fig. 3A). Upon stimulation with IL-1α no clear gradient between the three groups was observed (Supplementary Fig. 3B).

Reactive oxygens species (ROS) are known to be elevated in tumors and can lead to tumor progression[22,23]. The capacity of freshly isolated peripheral blood CD14+ CD16− classical monocytes to produce ROS was tested by stimulation with PMA or opsonized zymosan. No significant differences between the groups were observed. However, a subtle increasing gradient was observed in ROS production capacity from HC to MNG patients to TC patients (Fig. 3B).

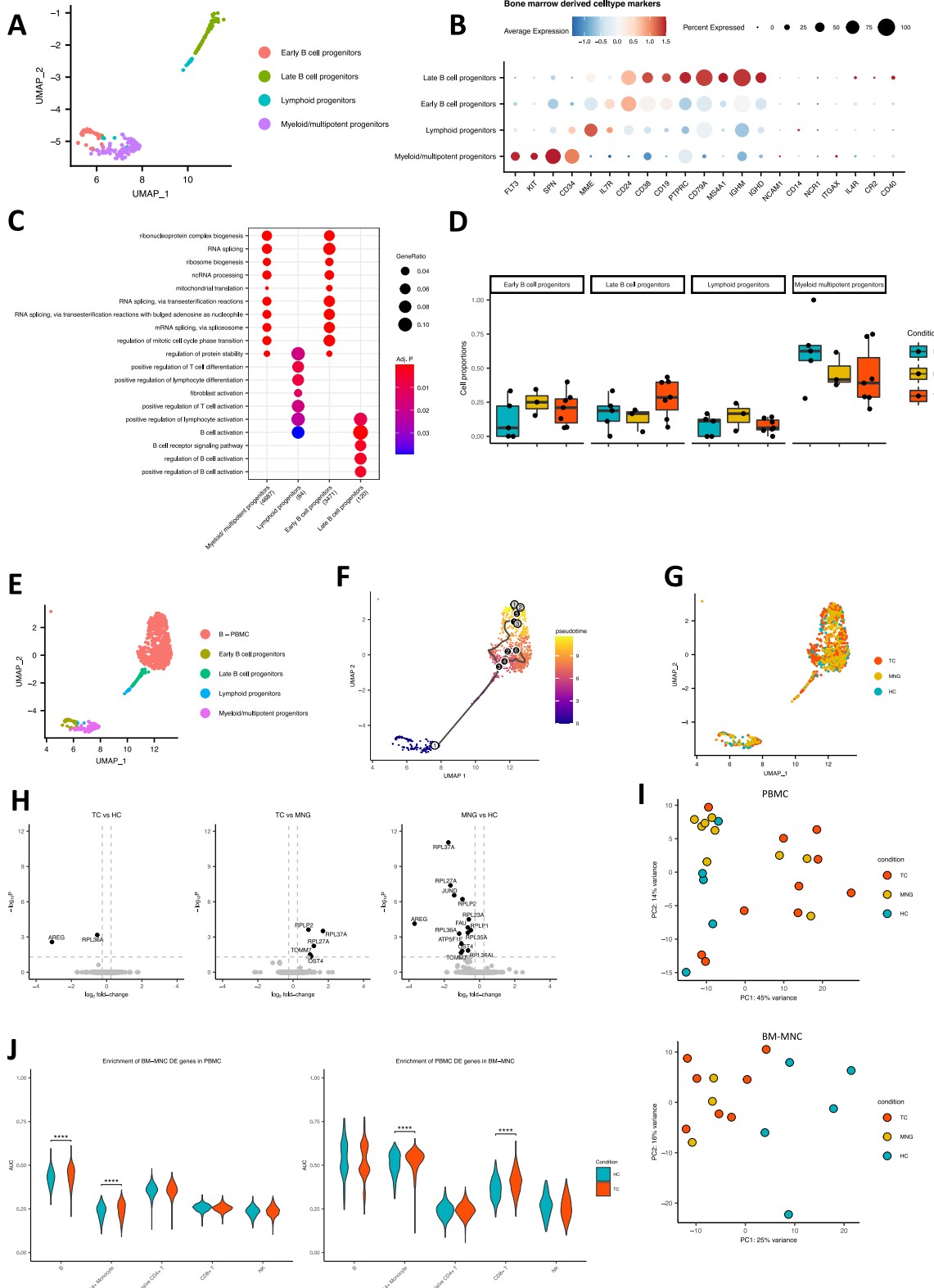

## Diminished amplification of cytokine production induced by cancer cells in myeloid cells of TC patients

After transwell co-culture of monocytes with the papillary TC cell line TPC-1 (see scheme in Fig. 3C) followed by LPS stimulation, monocytes of all groups showed a similar increase in TNF-α response (Fig. 3D, first graph). Furthermore, all groups showed an increase in IL-6 after stimulation, which was highly significant in the MNG group (Fig. 3D, third graph). Notably, the increase in IL-6 was the weakest in TC patients and the strongest in MNG patients. The increase in IL-6 response after co-culture was significantly lower in TC patients and HC compared to MNG patients. In contrast, IL-1β responses significantly decreased in TC patients and HC, but remained unchanged in MNG patients. (Fig. 3D, second graph). IL-1Ra responses showed no significant increases in TC patients and in HC whereas a significant increase was

**Fig. 2 | Single-Cell Transcriptomics of Bone Marrow-derived Mononuclear Cells.**
**A** UMAP showing 275 CD34$^+$ progenitor cells derived from bone marrow tissue from 15 samples (HC $n = 5$, MNG $n = 3$, TC $n = 7$). Canonical markers were used to annotate the different cell types (**B**). **B** Dot heatmap showing the expression of cell type markers used to annotate the bone marrow-derived progenitor cell types. **C** GO term enrichment analysis of the cell type-specific markers for each of the progenitor cell populations. Over-representation analysis was used to determine significance. $P$ values were measured using the hypergeometric distribution and adjusted using Benjamini-Hochberg correction. **D** Box plots of cell proportions within the CD34$^+$ progenitor cell populations derived from the bone marrow in healthy controls ($n = 5$), MNG ($n = 7$) and TC ($n = 3$). The center line per box plot refers to the median, bounds are the 25th and 75th percentiles (interquartile range; IQR). Box plot whiskers are the smallest and largest values no further than 1.5*IQR from the box plot bounds. No significant differences were found between the conditions. **E–G** UMAP of the CD34$^+$ progenitor cell populations, along with the PBMC-derived B cells. Colored by cell type, pseudo time and condition, respectively. **H** Volcano plots showing differential expression of (i) TC vs. HC, (ii) TC vs. MNG and (iii) MNG vs. HC within the myeloid multipotent progenitors. $P$ values were obtained from the MAST test (two-sided) and adjusted using Bonferroni correction. **I** PCA plots showing the first two principal components of the pseudo-bulk RNAseq profile from the periphery (PBMC) and the bone marrow-derived mononuclear cells (BM-MNC). **J** Violin plots showing enrichment of the PBMC cell type-specific differentially expressed genes in BM-MNC cell populations (left), and vice-versa (right). Wilcoxon two-sided test between TC and HC for each cell type separately with Benjamini-Hochberg correction over all cell types for PBMCs and BM-MNCs separately. PBMCs $p$ values <0.0001, BM-MNC $p$ values <0.00001. Source data are provided as a Source Data file.

observed in the MNG group (Fig. 3D, fourth graph). Overall, MNG patients showed a different interleukin response to co-culture than TC patients and healthy controls.

### Blood circulating inflammatory markers in TC and MNG patients are increased compared to HC

The tumor may influence the functional programming of myeloid cells and their progenitors by releasing mediators into the bloodstream. Circulating inflammatory proteins were determined in plasma from the three groups of volunteers. After employing a cut-off of ≥35% of values under the detection limit, 75 out of the 92 inflammatory markers were analyzed in the blood samples. Differential analysis of peripheral blood inflammatory proteins revealed four significantly increased biomarkers in TC patients compared to healthy controls (CCL20 ($P = 0.004$, FC = 1.94), CXCL9 ($P = 0.032$, FC = 1.63), IL-6 ($P = 0.031$, FC = 1.39), CDCP1 ($P = 0.037$, FC = 1.52)) and one significantly decreased marker (NT-3 ($P = 0.048$, FC = 0.79)) (Fig. 3E). Furthermore, eight mediators were significantly increased in MNG patients compared to healthy controls (CDCP1 ($P = 0.017$, FC = 1.55), EN-RAGE ($P = 0.017$, FC = 1.45), CCL3 ($P = 0.025$, FC = 1.49), CXCL9 ($P = 0.029$, FC = 1.58), FGF21 ($P = 0.037$, FC = 1.72), TNFRSF9 ($P = 0.043$, FC = 1.49), FGF19 ($P = 0.045$, FC = 1.63), IL-6 ($P = 0.045$, FC = 1.59)) (Fig. 3F). Only one protein was significantly decreased in TC compared to MNG patients (FGF19 ($P = 0.021$, FC = 0.61)) (Fig. 3G). Interestingly, of the four markers being significantly increased in TC patients compared to healthy controls, CCL20 was specifically increased in TC patients (Supplementary Fig. 3E). The other markers, CXCL9, CDCP1 and IL-6 were significantly increased to a similar extent in both patient groups relative to healthy controls (Fig. 3E, F and Supplementary Fig. 3E). This suggests comparable changes in the inflammatory phenotype of MNG and TC patients, but a different responsiveness to these changes by the myeloid cells of these two patient groups.

### Therapy effects on the phenotype and function of monocytes and circulating inflammatory markers

Next, patients were followed longitudinally before and at 3 time points during/after treatment according to standard of care, i.e., (hemi)thyroidectomy and/or $^{131}$I treatment (Fig. 4A) to assess the effect of treatment on circulating CD14$^+$ CD16$^-$ classical monocytes and inflammatory markers in blood. According to the clinical standard of care and the nature of the diseases, patients with TC received higher doses of $^{131}$I than MNG patients (Table 1). Blood was drawn at baseline, 30 days after surgery, 7 days after treatment with $^{131}$I and 30 days after $^{131}$I treatment.

Complete blood counts were measured in EDTA blood samples at all time points. Surgical removal of the thyroid gland did not affect cell counts, whereas treatment with $^{131}$I reduced the lymphocyte counts in both patient groups, showing a significant reduction at 7 days after $^{131}$I in the TC group ($p = 0.034$) and a significant reduction at 7 days ($p = 0.011$) and 30 days after $^{131}$I in patients with MNG ($p = 0.003$)

compared to baseline (Fig. 4B). This effect was also seen in the WBC count of MNG patients 30 days after $^{131}$I ($p = 0.009$).

### The impact of treatment on functional phenotype of peripheral blood monocytes

Functional analysis of circulating classical monocytes at these different time points following LPS stimulation, revealed that TNF-α, IL-1β, IL-1Ra and IL-6 production were all unaffected by (hemi)thyroidectomy or $^{131}$I treatment in TC and MNG patients (Fig. 4C). Similar effects were observed upon stimulation with P3C (Supplementary Fig. 3C) and IL-1α (Supplementary Fig. 3D).

ROS responses were not changed after (hemi)thyroidectomy in both patient groups. 7 days after $^{131}$I, there was a trend towards reduced ROS production decreased in the TC group after $^{131}$I treatment, both after PMA and zymosan stimulation. 30 days after $^{131}$I, ROS production also showed a decreasing trend in TC patients after stimulation with zymosan (Fig. 4D). Repeated two-way ANOVA test further confirmed similar patterns of the effect of treatment on immune cells numbers and function (Supplementary Fig. 4). To exclude a direct effect of iodine on cytokine and ROS production capacity, we stimulated monocytes with LPS (24 h for cytokine production) or zymosan (1 h for ROS production) in the presence of different concentrations of iodine (1 and 10 nM). Iodine did not influence production of cytokines or ROS when 1 and 10 nM concentrations were used, with only a moderate effect on TNF (Supplementary Fig. 5). These data argue that the changes of cytokine and ROS production after treatment of patients are most likely due to the removal of the tumor (and thus tumor-associated mediators), rather than the effects of iodine treatment itself.

## Discussion

An increasing amount of evidence shows the important role of the innate immune system in general, and of myeloid cells in particular, in cancer. Most knowledge so far has been gathered on innate immune cells infiltrating in the tumor. However, tumors can have systemic effects by factors such as secreted proteins, metabolites, exosomes or even circulating neoplastic cells, which may affect innate immune cells in the circulation or even their progenitors in the bone marrow. Therefore, we analyzed to what extent circulating monocytes and myeloid progenitor cells in the bone marrow may already be affected by the presence of a TC tumor in situ. Furthermore, we hypothesized that the effective treatment of TC using surgery and radioactive iodine in the patients may exert immunomodulatory effects on circulating monocytes. In line with our hypothesis, the bone marrow progenitors are indeed affected in tumor patients, as TC patients show a different signature in the bone marrow progenitor composition. Furthermore, transcriptional differences are observed in the differentiated immune cells in the bone marrow compartment. In the circulation, classical monocytes from TC patients show features of a more tolerant functional phenotype and transcriptional program. Interestingly, in their

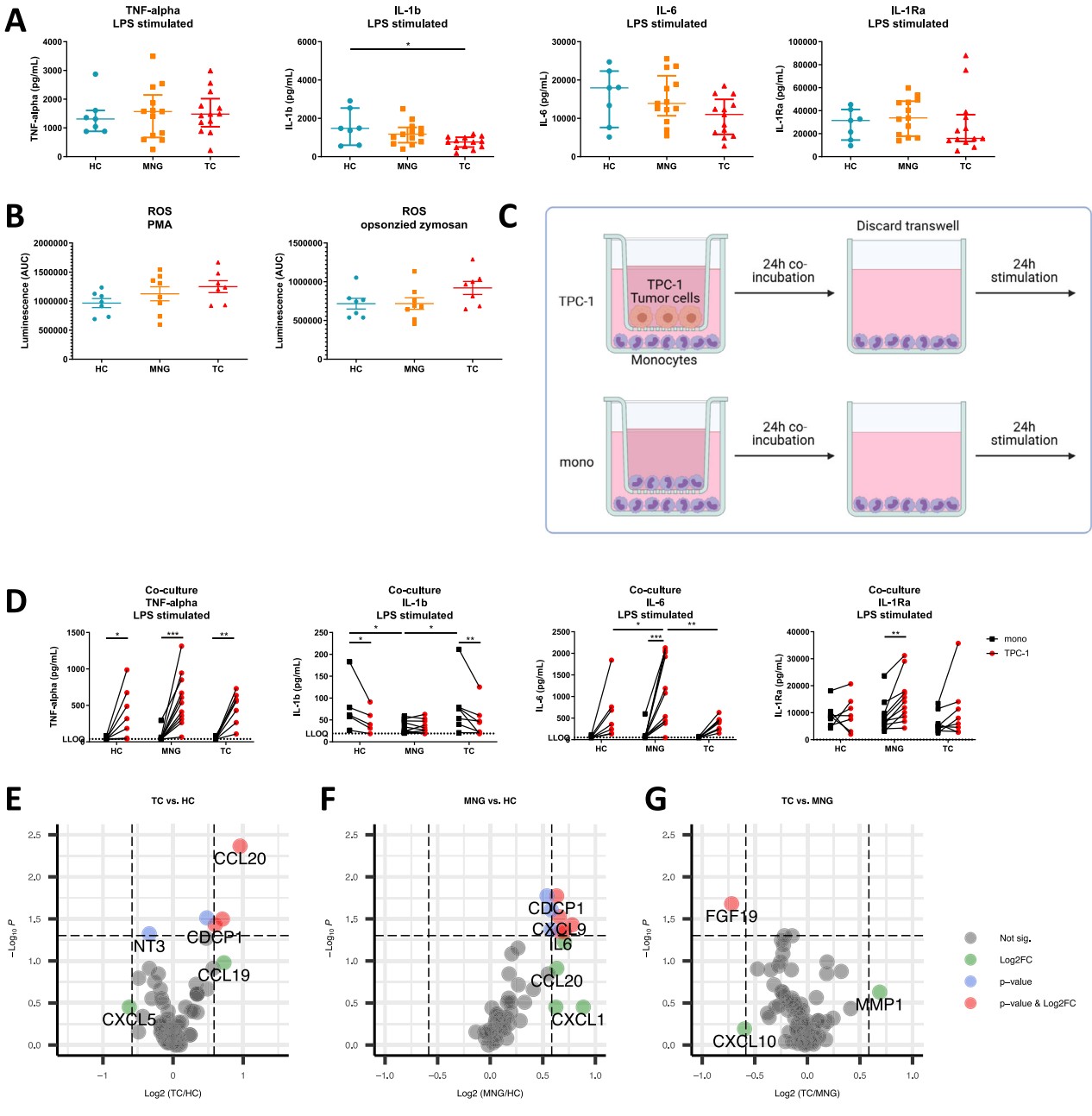

**Fig. 3 | Functional phenotype of peripheral blood classical monocytes and circulating inflammatory markers. A** Cytokine release of CD14⁺ CD16⁻ classical monocytes after 24 h stimulation with TLR4 ligand LPS. (HC $n = 7$, MNG $n = 13$, TC $n = 13$) Median with IQR. Corrected for age and sex. **B** ROS production of CD14⁺ CD16⁻ classical monocytes during 1 h stimulation with PMA or opsonized zymosan. (HC $n = 7$, MNG $n = 8$, TC $n = 7$) Mean ± SEM. Corrected for age and sex. One-way ANOVA followed by Bonferroni correction. **C** Graphical illustration of co-culture experiment. CD14⁺ CD16⁻ classical monocytes were indirectly co-cultured with TPC-1 cells by using a transwell system. Monocytes in the transwell were used as a control condition. After 24 h co-incubation the transwell with the tumor cells was discarded and the monocytes were stimulated with LPS for 24 h. Illustration created with BioRender.com. **D** Cytokine production of CD14⁺ CD16⁻ classical monocytes after indirect co-culture with TPC-1 cells followed by 24 h LPS stimulation. (HC $n = 7$,

MNG $n = 11$, TC $n = 8$) Paired data are connected by lines. Repeated measures two-way ANOVA followed by Bonferroni correction. Adjusted $p$ values: TNFa HC $p = 0.021$, MNG $p < 0.001$, TC $p = 0.004$; IL-1b HC $p = 0.014$, TC $p = 0.006$, mono HC vs. MNG $p = 0.043$, MNG vs. TC $p = 0.041$; IL-6 MNG $p < 0.001$, TPC-1 HC vs. MNG $p = 0.047$, TC vs. MNG $p = 0.002$; IL-1Ra MNG $p = 0.009$. **E**–**G** Comparison of circulating inflammatory mediators in peripheral blood plasma between groups. Volcano plots of fold change and $p$ values for TC versus HC, MNG versus HC and TC versus MNG, respectively. Red circles (fold change >1.5 or <0.5 and $p$ value <0.5) and blue circles ($p$ value <0.5) show significantly enriched or depleted inflammatory markers. Green circles (fold change >1.5 or <0.5) show markers enriched or depleted, without reaching significance. Source data are provided as a Source Data file.

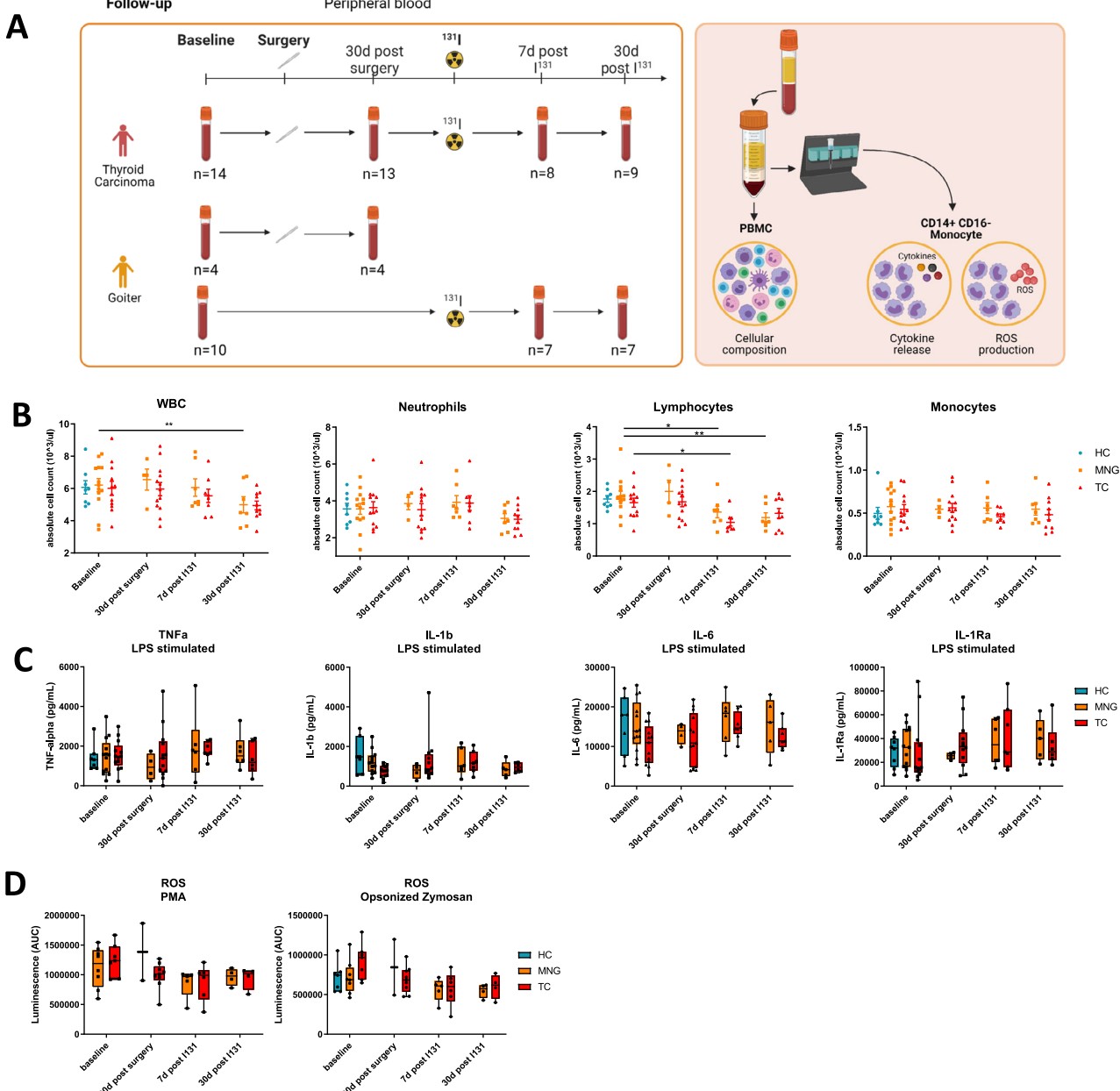

**Fig. 4 | Effect of treatment on functional phenotype of peripheral blood classical monocytes and circulating inflammatory mediators in peripheral blood plasma. A** Study Design Follow-up Measurements: Peripheral blood was collected from TC and MNG patients before treatment (baseline), 30 days after surgery, 7 days and 30 days after treatment with radioactive iodine. Illustration created with BioRender.com. **B** Differential cell counts in whole blood. (HC $n = 8$, MNG $n = 4$–13, TC $n = 8$–13). Mean ± SEM. Mixed effects model (repeated measures two-way ANOVA with missing values) followed by Bonferroni correction. Each time point was compared to the baseline time point, as not all study participants received both treatments. Adjusted $p$ values: MNG baseline vs. 7d post I131 $p = 0.011$, baseline vs. 30d post I131 $p = 0.003$; TC baseline vs. 7d post I131 $p = 0.034$. **C** Cytokine release of CD14[+] CD16[-] classical monocytes after 24 h stimulation with TLR4 ligand LPS. (HC

$n = 7$; MNG: baseline $n = 13$, 30d post-surgery $n = 4$, 7d post I131 $n = 6$, 30d post I131 $n = 5$; TC: baseline $n = 13$, 30d post-surgery $n = 13$, 7d post I131 $n = 8$, 30d post I131 $n = 7$). Median with IQR and Tukey whiskers. Corrected for age and sex. Mixed effects model (repeated measures two-way ANOVA with missing values) followed by Bonferroni correction. Each time point was compared to the baseline time point, as not all study participants received both treatments. **D** ROS production of CD14[+] CD16[-] classical monocytes after 24 h stimulation with TLR4 ligand LPS. (HC $n = 7$, MNG: baseline $n = 8$, 30d post-surgery $n = 2$, 7d post I131 $n = 5$, 30d post I131 $n = 4$; TC: baseline $n = 7$, 30d post-surgery $n = 9$, 7d post I131 $n = 6$, 30d post I131 $n = 4$) Median with IQR and Tukey whiskers. Two-way ANOVA followed by Bonferroni correction. Corrected for age and sex. Source data are provided as a Source Data file.

function, the circulating classical monocytes show a gradient from healthy volunteers to benign MNG to malignant TC. After treatment with [131]I, the overproduction of ROS production by peripheral blood classical monocytes is partly restored towards normal levels. Our most important findings of this study are summarized in Fig. 5A, B.

Elevated peripheral blood monocyte counts have been associated with worse disease prognosis in several cancer types[24–29], and in

anaplastic TC a low lymphocyte-to-monocyte ratio has been associated with poor overall survival[30]. The elevated monocyte numbers can be either caused by an increased monopoiesis or an enhanced mobilization from the bone marrow by CCL2, which is often elevated in serum of cancer patients[31–34], arguing for the involvement of the bone marrow compartment in cancer patients. Based on these earlier studies, we hypothesized that soluble tumor-derived factors influence the

**Fig. 5 | Graphical summary of main findings. A** Baseline. **B** After treatment with radioactive treatment. Created with BioRender.com.

developmental trajectory of bone marrow immune cell progenitors: practically, the immunosuppressive phenotype of myeloid cells in cancer is determined at a much earlier stage than currently thought. This hypothesis is supported by the current data in TC patients even without tumor metastasis or tumor infiltration in the bone marrow, in whom important changes are already present in immune cell progenitors in the bone marrow.

Interestingly, we show that cell proportion analysis of bone marrow-derived progenitor cell types revealed a decrease in myeloid/multipotent progenitor cell proportion for TC samples, although the

differences did not reach statistical significance due to the relatively low number of samples available. This trend is reversed for the early and late B-cell progenitors for which the TC samples have the highest proportions. This may indicate a shift in TC patients from myelopoiesis towards lymphopoiesis, and specifically B-cell progenitors.

Previous studies investigating the immune cells phenotype in cancer patients have compared this phenotype to that of healthy controls. In the present study we compared the immune cells of patients with TC with those collected from patients with MNG, which provides an in vivo model of a benign thyroid condition. Our study now demonstrates, that while some of these phenotypic changes are specific for malignant processes, many others are not and could rather reflect reactive changes to a non-malignant pathogenic condition. Interestingly, large transcriptional differences were observed in the CD4+ T-cell population from peripheral blood that mostly distinguish the MNG samples specifically. Furthermore, in classical monocytes from TC patients the inflammatory phenotype was reduced to some extent with a gradient from healthy controls to MNG to TC. On the other hand, circulating CD14+ monocytes from TC patients, as well as MNG patients, show upregulation of genes responsible for antigen presentation at the transcriptional level compared to healthy controls. This suggests that the MNG group can be considered as biologically unique entity. Different etiological factors are involved in the pathogenesis of MNG and in TC and although MNG is also a neoplastic condition, it cannot automatically be regarded as a precancerous condition or an intermediate state towards cancer development. The observed gradient in phenotype of immune cells could be related to the neoplastic state of MNG, but other non-specific factors cannot be excluded.

In TC patients we observed changes in the transcriptional programs of the immune cell progenitors in the bone marrow that also reflect the general effect observed on the balance between lympopoiesis and myelopoiesis. In this respect, pathway analysis of RNA-sequencing data demonstrated activation of lymphocyte differentiation pathways in both lymphoid cell progenitors and late B-cell progenitors. In contrast, myeloid multipotent progenitors display increases in pathways influencing RNA splicing and ribosome biogenesis, for which future studies should investigate the biological relevance. Moreover, the shared signature between CD14+ monocytes in the bone marrow and the circulation suggest that CD14+ monocytes in the bone marrow have already changed their transcriptional program before entering the circulation. Future therapeutic approaches against immunosuppressive CD14+ monocytes thus need to also target the bone marrow-resident cells in order to reach full efficacy. We aimed to functionally validate these findings by assessing cytokine production capacity of monocytes in the circulation: it is well known that monocytes have a short lifetime in the blood of only a few days, and they likely mirror changes that they had undergone a few days earlier in the bone marrow. A reduced responsiveness to stimulation of peripheral monocytes observed in TC patients is in line with earlier findings in breast cancer patients, showing an impaired production of IL-1β, IL-6 and TNFα in response to LPS[35,36]. A reduced cytokine production capacity of monocytes from TC patients may possibly be induced by circulating tumor-derived mediators that change their function or that of their precursors in the bone marrow: this hypothesis needs to be investigated in future studies. Moreover, upon exposure to factors secreted by a TC cell line in a transwell model ex-vivo, classical monocytes from TC patients show a strongly impaired secretion of IL-6 compared to cells isolated from MNG patients, confirming the presence of a suppressed cytokine production capacity in response to tumor-derived factors. This might indicate a decreased ability of monocytes from TC patients to respond efficiently after the encounter with tumor cells.

[131]I is an effective therapy in patients with TC, but patients with advanced or metastatic disease may become resistant to this

treatment. Tumor ablation by ionizing radiation is known to result in DNA damage leading to activation of the cGAS-STING pathway and thereby a type I interferon response[37,38]. Furthermore, radiation leads to release of danger signals and tumor antigens. These effects can stimulate immune cells and promote systemic anti-tumor immune responses in patients. TAMs can be activated by ionizing radiation of the tumor as well, leading to enhanced phagocytosis and proinflammatory cytokine expression[39]. Therefore, we assessed whether [131]I treatment in TC patients has an immunomodulatory effect on circulating monocytes. Similar to patients with TC, some patients with MNG undergo surgical removal of the thyroid and some undergo treatment with [131]I. This provided the opportunity to investigate the effect of different treatments on the immune cell phenotype and help identify cancer-specific changes.

We observed that [131]I treatment significantly reduces the lymphocyte counts in both patient groups. This finding confirms previous observations showing lymphopenia after [131]I treatment in TC patients[40–43]. Bone marrow suppression is a well-known side-effect of [131]I treatment, which generally recovers after 6 months[40]. We confirm that this is a general side-effect of the therapy, as we also observe it in patients with MNG, although these patients receive a lower dose of [131]I. Interestingly, a TC-specific effect of tumor ablation is observed in the circulating levels of several inflammatory markers.

However, the cytokine production capacity of peripheral blood classical monocytes did not show any differences after treatment in TC and MNG patients. Interestingly, after treatment with [131]I, ROS production of circulating classical monocytes in TC patients decreased to the levels similar to those found in the healthy volunteers. High ROS levels in a tumor play a role in the early phase of tumorigenesis, as they are actively mutagenic for local neoplastic cells by inducing DNA damage and genomic instability and thereby accelerating their malignant transformation[44]. One may hypothesize that one mechanism through which [131]I treatment may reduce tumorigenesis is by reducing ROS production from tumor-infiltrating monocytes. The functional changes we observe in circulating monocytes after [131]I treatment suggest that the tumor-induced systemic effects can be reversed by the treatment. In order to assess whether this is induced by a direct effect of [131]I on myelopoiesis we investigated the effect of iodine on cytokine and ROS production in an ex vivo model. While an in vitro assay has limitations, this argues that the changes of immune function after treatment of patients are most likely due to the removal of the tumor and tumor-associated mediators, rather than the effects of [131]I treatment itself. Nonetheless, we cannot completely exclude that the effect observed after [131]I treatment may be due to the beta radiation, as most of the patients did not have large amounts of cancer cells remaining after the primary surgery.

This study also has limitations. First, our study is performed with a relatively small sample size, due to the invasive character of a bone marrow aspiration from this fragile patient population, and TC being a relatively rare type of cancer. Second, this study involved only two patients with metastasized disease. In the future, performing a similar study in patients with more advanced disease, such as metastasized, poorly differentiated, anaplastic, or [131]I -refractory TC, would be very insightful. Third, it would have been ideal to be able to study bone marrow function longitudinally in time during the disease and treatment, but this is obviously not possible due to ethical considerations. Finally, quality and reliability of our pseudo-bulk analyses rely on the underlying single-cell results. While we have not observed batch effects of quality artefacts, pseudo-bulk analyses could be influenced by technical factors such as specific cell type losses.

In conclusion, this study shows that changes in the number and phenotype of myeloid cells, key players of the innate immune system, start in TC patients long before the cells infiltrate the tumor, possibly already at the level of the bone marrow. Our findings underline the important potential role of innate immune cells in cancer and cancer

treatment, such as [131]I. Further exploration of the modulation of the innate immune system in cancer, for example by intersecting the results presented here with sophisticated drug-target networks[45] (Supplementary Data 2), may help to develop novel therapeutic targets in the future that will complement immune checkpoint inhibitors and conventional treatments regimens.

## Methods

### Patient selection
Peripheral blood and bone marrow were collected from HC and consecutive patients with MNG or newly diagnosed TC visiting our center between March 2018 and November 2019. Criteria for exclusion were age <18 years, mental incompetence, pregnancy or breastfeeding, inflammatory or infectious diseases or an immunosuppressive status, use of medication interfering with the immune system (e.g., glucocorticosteroids, nonsteroidal anti-inflammatory drugs, cytostatics) diagnosis of clinical thyroid dysfunction at time of inclusion, reduced platelet counts or other conditions associated with an increased risk of bleeding, severe comorbidities such as other active malignancies (except for basal cell carcinoma), serious psychiatric pathology and a self-reported alcohol consumption of >21 units per week. The study was approved by the local Ethics Committee (Ethical approval CMO Arnhem-Nijmegen 2017-3628) and registered at ClinicalTrials.gov (NCT03397238). All donors provided written informed consent.

### Blood sampling and bone marrow aspiration
Blood was collected by venipuncture in ethylenediaminetetraacetic acid (EDTA) blood collection tubes for all study participants at baseline. In patients with MNG end TC blood was also collected at different time points following standardized treatment regimens 30 days post-surgery and/or 7- and 30 days post [131]I (see overview in Fig. 4A). Bone marrow was aspirated in participants undergoing surgery from the posterior iliac crest or sternum in syringes containing sodium heparin (3:1 ratio) according to standard practice by an experienced physician assistant. The aspiration was performed immediately after induction of general anesthesia and tracheal intubation, to limit the time of exposure of anesthetic agents. Also, prophylactic intravenous administration of dexamethasone for postoperative nausea was not given before bone marrow was aspirated. Plasma was obtained from EDTA blood by centrifugation and stored at −80 °C until further use.

### Isolation of PBMCs and monocytes
Prior to mononuclear cell (MNC) enrichment, the bone marrow aspirate was filtered and washed with Phosphate Buffered Saline (PBS, Braun Melsungen, Germany). Thereafter, the same procedures were followed for peripheral blood MNCs (PBMCs) and bone marrow mononuclear cells (BM-MNCs). After removal of platelets by low-speed centrifugation (200 × g, 20 min) PBMCs/BM-MNCs were isolated by density gradient centrifugation using Ficoll-plaque (GE Healthcare, Diegem Belgium) and SepMate™ tubes (Stemcell Technologies, Vancouver, Canada). PBMCs and BM-MNCs were washed twice in cold PBS (Braun Melsungen) and resuspended in low glucose culture medium: RPMI 1640, no glucose culture medium (Life Technologies, Carlsbad, CA, USA) supplemented with gentamycin 50 μg/ml, pyruvate 1 mM, glutamine 2 mM, glucose 5 mM and HEPES 10 mM (Life Technologies). Before and after each isolation step, cells were counted using the Sysmex-XN 450 hematology analyzer (Sysmex, Norderstedt, Germany). 20–40 × 10⁶ cells were cryopreserved in CryoStor CS10 medium (Stemcell Technologies) until single-cell RNAseq analysis.

Classical monocytes (CD16⁻ CD14⁺) were isolated within the PBMC fraction by depletion of CD16⁺ cells using CD16 microbeads (Miltenyi Biotec, Bergisch Gladbach, Germany) followed by positive selection of CD14⁺ cells with CD14 microbeads (Miltenyi Biotec) according to the manufacturer's instructions.

## Flow cytometry

Cell populations in BM-MNCs were identified with the FACSAria III (Becton Dickinson, Franklin Lakes, NJ, USA). $10 \times 10^6$ cells were washed in PBA (0.5% BSA in PBS) after thawing, blocked with Fc-block (Miltenyi Biotec, Bergisch Gladbach, Germany), stained for 45 min at 4 °C in the dark with the monoclonal antibodies (CD34 BV421 Biolegend # 343609, CD10 APC Biolegend; #312209, CD110 PE/TexasRed BD Biosciences; #562416, CD45Ra FITC Biolegend; #304105, CD90 PE Biolegend; #328109, CD45 PerCP Biolegend; #304025, CD123 BV785 Biolegend; #306031, CD38 BV510 Biolegend; #356611, CD3 APC/Cy7 Biolegend; #317341, CD20 APC/Cy7 Biolegend; #302313, CD19 APC/Cy7 Biolegend; #363009, CD15 APC/Cy7 Biolegend; #323041) and washed with PBA. Samples were acquired using BD FACS ARIAIII (BD Biosciences) and the FACS Diva software (BD Biosciences, USA). Data was analyzed using FlowJo V10 (Tree Star, USA) and is presented as percentage of live CD45+Lin− CD34+.

## DNA genotyping

DNA was isolated from 0.5 ml whole blood with the ReliaPrep chemistry (Promega) automated on the Freedom Evo Robot (Tecan). Genotyping was performed using the Illumina Infinium Global Screening Array (GSA)-MDv3 array.

## Single-cell RNA-sequencing

Frozen PBMCs and BM-MNCs were thawed according to the instructions given by 10x Genomics in preparation for their Single-Cell protocols (Fresh frozen human peripheral blood mononuclear cells for single-cell RNA-sequencing protocol- Document CG00039). Briefly, frozen PBMCs and BM-MNCs were recovered by thawing rapidly at 37 °C and immediately performing sequential dilution (1:1) with pre-warmed RPMI medium containing 40% FBS. The dilution step was repeated every minute for a total of 5 times, followed by centrifugation at $300 \times g$ for 5 min. After centrifugation, cells were resuspended in PBS with 0.04% BSA, passed through a 40μm cell strainer and cell concentrations and viability were determined using an automated Cell counter (TC20 automated Cell counter, Bio-rad). All samples had a minimum viability of ≥ 70%. Next, samples were prepared for scRNA-seq according to the 10X Genomics Chromium™ Single-Cell 3′ v2 RNA-sequencing specification. Cells from 5-6 samples were pooled for one reaction containing the recommended range of cells (700-1200 cells/ul) and loaded at a volume with a targeted cell recovery count of 1000 cells for each sample. The generated cDNA was used for Illumina next-generation sequencing using a NextSeq500-v2 150 cycle kit with a sequencing depth of 25.000 reads/cell.

Reads from scRNA-seq were aligned to the GRCh38 human genome using *CellRanger* (v3.1.0, 10X genomics) to generate a count matrix recording the number of transcripts (UMIs) for each gene in each cell, per pool. Demultiplexing of the pooled samples was performed using *souporcell*, a genotype-free clustering method[46]. We matched SNPs called from each cluster in the pool to the individual genotypes to identify the cells belonging to each donor. We did not observe any significant batch effects between the different sequencing pools.

## Stimulation experiments

Per well, 200.000 CD16− CD14+ monocytes were stimulated in duplicate in flat-bottom 96-well plates for 24 h with low glucose culture medium, 10 ng/ml LPS (*Escherichia coli* strain O55:B5, Sigma Chemical Co, St. Louis, MO), 10ug/ml Pam3CysK4 (P3C) (EMC Microcollections, Tübingen, Germany), and recombinant human IL-1a (200-LA) (R&D, Minneapolis, MN, USA). After plate centrifugation, supernatants were stored at −80 °C until cytokine assessment. The effect of iodine on cytokine production capacity was assessed by stimulating monocytes with LPS in the presence of different concentrations of iodine (1, 10 and 50 nM), with cytokines being measured 24 h later.

## ROS assay

Per well, 100.000 CD16− CD14+ monocytes were stimulated in triplicate in white non-transparent flat-bottom 96-well plates with low glucose culture medium, 50 ng/ml Phorbol 12-myristate 13-acetate (PMA) (Sigma) or 3 mg/ml of plasma-opsonized Zymosan A (Sigma). ROS formation was detected by a chemiluminescence assay using 0.1 mM 5-amino-2,3-dihydro-1,4-phthalazinedione (luminol) (Sigma). The luminometer measured chemiluminescence in the integration mode at 37 °C every 142 s for 1 h after luminol had been added. The effect of iodine on ROS production capacity was assessed by stimulating monocytes with zymosan in the presence of different concentrations of iodine (1, 10 and 50 nM), with ROS being measured during 1 h of stimulation.

## Co-culture model

Co-culture experiments were performed using the TC cell line TPC-1 (papillary, RET/ PTC rearrangement). Cancer cells were cultured in RPMI 1640 Dutch modification (Life Technologies) supplemented with gentamycin 50ug/ml, pyruvate 1 mM, glutamax 2 mM and 10% fetal calf serum (Life Technologies). A transwell system used ThinCert™ cell culture inserts on a 24-well plate (Greiner Bio-One GmbH, Austria). A total of 50.000 tumor cells were seeded in the transwell 24 h prior to the start of the co-culture to attach, while medium was added to the lower compartment. After 24 h incubation at 37 °C, 5% $CO_2$, the tumor cells were washed with PBS (Braun Melsungen). A total of 500.000 classical monocytes in 500 μl low glucose culture medium was added to the lower compartment of the transwell system and incubated with the tumor cells in low glucose culture medium in the upper compartments. As a control, monocytes were added to the upper compartment instead of tumor cells.

After 24 h incubation, the cell culture inserts containing TPC-1 cells or control monocytes were discarded and the (tumor-induced) monocytes in the wells were stimulated for 24 h with 10 ng/ml LPS (*E. coli* strain O55:B5, Sigma-Aldrich) or RPMI medium as control. At the end of the incubation period, supernatant was collected and stored at −80 °C until cytokine assessment.

## Cytokine assays

Interleukin-6 (IL-6), tumor necrosis factor α (TNF-α), IL-1Ra and IL-1β concentrations were determined in supernatants by commercial enzyme-linked immunosorbent assay (ELISA) kits according to the instruction of the manufacturer (R&D, the Netherlands). To minimize batch effects during measurement, all time points belonging to one patient were measured on the same plate.

## Proximity extension assay (PEA)

Circulating inflammation biomarkers were analyzed by the analysis service of Olink Proteomics AB (Uppsala, Sweden) using their PEA based Target 96 Inflammation panel[47]. This analysis simultaneously measured 92 selected inflammatory proteins in plasma. For each protein, there are two antibodies with unique DNA oligonucleotides. As the two antibodies come into proximity, their DNA oligonucleotide hybridize. This DNA is extended, creating a double-stranded DNA barcode, which is protein-specific. The reporter DNA strands are then quantified using qPCR. Four internal controls and two external controls were included in the assay. The raw targeted proteomic data is supplied in Supplementary Data 1.

## Statistical analysis

**Patient characteristics.** 1-way ANOVA or Chi-square were used to determine differences between the three groups. Student's t-test was performed to compare TC and MNG group. Analyses were performed in Graphpad Prism 5 (CA, USA).

**Cell counts**. Cell counts were normally distributed and differences between groups at baseline were analyzed using one-way ANOVA test followed by Bonferroni correction. For the longitudinal analysis, a mixed effects model (repeated measures two-way ANOVA with missing values) followed by Bonferroni correction was performed. Each time point was compared to the baseline time point, as not all study participants received both treatments. Analyses were performed in Graphpad Prism 8 (CA, USA). Data are shown as mean ± SEM

**scRNA-seq data analysis**. We applied three thresholds to exclude low quality cells, only cells with less than 20% mitochondrial genes and between 250 and 6000 unique genes were included. Gene expression data was normalized by total expression per cell, multiplied by 10,000 and subsequently log-transformed. Finally, gene expression values were scaled and the percentage of mitochondrial and ribosomal genes regressed out. Using Seurat v4.0.1[48] in R v4.0.3 the top 2000 variable genes were selected and used to calculate 20 principal components (PCs). These 20 PCs were then used together with the RunUMAP, FindNeighbors, and FindClusters functions of Seurat to cluster the cells in an unsupervised manner. These clusters were then annotated with their respective cell types using known marker gene expression. Similarly, we re-clustered cells originating in one bone marrow-derived cell cluster expressing progenitor specific marker genes and we annotated them into more specific cell subtypes using known marker genes. We calculated differential expressed genes (DEGs) between conditions using the FindMarkers function from Seurat. We specifically used the MAST[49] test while correcting for sample age and sex. *P* values were adjusted using Bonferroni correction and a log fold change threshold of 0.25 was applied. We then used the resulting differentially expressed genes as input for KEGG pathway enrichment and GO term enrichment using the R package *ClusterProfiler*[50]. Only pathways with at least 5 overlapping genes were retained. The GeneRatio represents the number of input genes annotated a specific GO term, divided by the total number of genes annotated to this term.

We analyzed cell proportion differences by applying Dirichlet regression[51] using all cells, exclusively PBMC-derived cells, bone marrow-derived cells and progenitor cell types separately. The Dirichlet model was fitted on the cell proportions for each cell type, per comparison of two conditions separately (TC vs. HC, TC vs. MNG and MNG vs. HC). *P* values were corrected for multiple testing using Bonferroni correction across all cell types and comparisons of conditions.

Cell type transition using each cell's pseudo-time was estimated using Monocle3 v0.2.3[20]. We first selected the unsupervised cell cluster expressing progenitor cell type markers and due to the high expression of B-cell progenitor markers in some of these bone marrow-derived progenitor cells we included the PBMC-derived B cells for this pseudo-time trajectory analysis as well.

We calculated differentially expressed genes within PBMCs and BM-MNCs, separately for each cell type comparing thyroid-cancer versus healthy controls. In line with our previous analyses, genes significantly different after multiple testing were retained, and no fold-change filter was applied. Subsequently, we employed the R package AUcell[52] to calculate the enrichment of the PBMC differentially expressed genes in the corresponding BM-MNC cell type, and vice-versa. For visualization purposes, we performed pseudo-bulk analysis within PBMCs and BM-MNCs, respectively. To test whether HC were separated from patients (TC and MNG) combined, we fitted a logistic regression model: *condition -PC* for PC1 and PC2.

**Cytokine assays**. Cytokine values were non-normally distributed, as previously demonstrated in detail[53]. Data was corrected for age and sex. Differences in cytokine production at baseline were analyzed using Kruskal–Wallis test followed by Dunn's multiple-comparison test. Data are shown as median with IQR. For the longitudinal analysis,

a mixed effects model (repeated measures two-way ANOVA with missing values) followed by Bonferroni correction was performed. Each time point was compared to the baseline time point, as not all study participants received both treatmentsData are shown as box plots with whiskers from min to max. In the co-culture experiments, paired data are analyzed by repeated measures ANOVA followed by Bonferroni correction. All cytokine analyses were performed in Graphpad Prism 8 (CA, USA).

**ROS assay**. ROS production was calculated as area under the curve during a measurement of 1 h. Data was corrected for age and sex by fitting a linear model that included age and sex as covariates. The residuals from this model were extracted and used for further statistical analyses. Luminescence was normally distributed and differences between groups at baseline were analyzed using one-way ANOVA test followed by Bonferroni. Data are shown as mean ± SEM. For the longitudinal analysis, a mixed model ANOVA followed by Bonferroni was performed, as due to small sample sizes and missing data a mixed model was not feasibleAnalyses were performed in Graphpad Prism 8 (CA, USA). Data are shown as box plot with whiskers from min to max.

**PEA analysis**. Samples that did not pass quality control and markers showing ≥35% of values below the limit of detection (LOD) in all groups of one sample type were excluded. The Normalized Protein Expression (NPX) values are expressed on a log2 scale and were linearized for fold-change calculations. Outliers were removed after Grubbs' test. All computational analyses were performed in R 4.0.3. Protein levels were compared between groups at baseline and within groups over time using Kruskal–Wallis test followed by post hoc Mann–Whitney *U* test. Packages used for the analyses included "ggplot2" and "ggpubr".

### Reporting summary
Further information on research design is available in the Nature Research Reporting Summary linked to this article.

## Data availability
The single-cell transcriptomic and genotypic data generated in this study have been deposited at the European Genome-Phenome archive (EGA) under accession number EGAS00001005594. These data are available under restricted access; access can be obtained upon request. Targeted proteomics data generated in this study are provided as Supplementary Data. Source data are provided with this paper.

## Code availability
Code generated to processed the single-cell RNA-sequencing data are freely available on Github or [https://github.com/CiiM-Bioinformatics-group/thyroid-cancer] or Zenodo. [https://doi.org/10.5281/zenodo.6973914].

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

## Acknowledgements

The authors are grateful to all volunteers for their participation in this study. The authors thank the Flow Cytometry Core Facility of Leiden University Medical Center for their support with flow cytometric measurements and the Human Genomics department of Leiden University Medical Center for their support with 10X single-cell assays. RTNM was supported by a grant (#10559) from the Dutch Cancer Society. K.R. was supported by a Radboud University Medical Center Junior researcher (Ph.D.) grant. Y.L. was supported by an ERC starting Grant (948207) and a Radboud University Medical Center Hypatia Grant (2018). C.J.X. was supported by Helmholtz Initiative and Networking Fund (1800167).

## Author contributions

Conceptualization and design: R.T.N.M., K.R., I.G., W.J.M.M., G.J.A., M.G.N., J.W.A.S., Y.L., C.J.X. Data analysis: K.R., M.Z., I.G., Y.K., M.J., Y.L., C.J.X. Patient recruitment, collection of biological material and experimental work: K.R., I.G., M.B., M.J., B.Z., W.H., H.J.B., J.H.W.W., M.J.R.J., L.A.M.C., I.C.H.E.G., R.T.N.M. Writing—initial draft: K.R., M.Z., I.G., Y.K., G.J.A., M.G.N., Y.L., C.J.X., R.T.N.M. Writing—review and editing: All co-authors.

## Competing interests

The authors declare no competing interests.

## Additional information

[1]Department of Internal Medicine, Radboud University Medical Center, Nijmegen, The Netherlands. [2]Radiotherapy and OncoImmunology Laboratory, Department of Radiation Oncology, Radboud University Medical Center, Nijmegen, The Netherlands. [3]Radboud Institute for Molecular Life Sciences, Radboud University Medical Center, Nijmegen, Netherlands. [4]Department of Computational Biology for Individualised Infection Medicine, Centre for Individualised Infection Medicine (CiiM), a joint venture between the Helmholtz Centre for Infection Research (HZI) and the Hannover Medical School (MHH), Hannover, Germany. [5]TWINCORE, a joint venture between the Helmholtz Centre for Infection Research (HZI) and the Hannover Medical School (MHH), Hannover, Germany. [6]Department of Haematology, Radboud University Medical Center, Nijmegen, The Netherlands. [7]Department of Laboratory Medicine, Laboratory of Hematology, Radboud Institute for Molecular Life Sciences, Radboud University Medical Center, Nijmegen, The Netherlands. [8]Department of Surgery, Radboud University Nijmegen Medical Centre, Nijmegen, The Netherlands. [9]Department of Radiology and Nuclear Medicine, Radboud University Nijmegen Medical Centre, Nijmegen, The Netherlands. [10]Department of Pathology, Radboud University Nijmegen Medical Centre, Nijmegen, The Netherlands. [11]Biomedical Engineering and Imaging Institute, Icahn School of Medicine at Mount Sinai, New York, USA. [12]Department of Medical Biochemistry, Amsterdam University Medical Centers, Amsterdam, The Netherlands. [13]Department of Biochemical Engineering, Laboratory of Chemical Biology, Eindhoven University of Technology, Eindhoven, The Netherlands. [14]Department of Internal Medicine, Division of Endocrinology, Radboud University Medical Center, Nijmegen, The Netherlands. [15]Department of Genomics and Immunoregulation, Life and Medical Sciences Institute, University of Bonn, Bonn, Germany. [16]These authors contributed equally: Katrin Rabold, Martijn Zoodsma. ✉e-mail: xu.chengjian@mh-hannover.de; romana.netea-maier@radboudumc.nl

