## [Peer Review File · Nature Communications]

Reprogramming of myeloid cells and their progenitors in patients with non-medullary thyroid carcinomaREVIEWER COMMENTS

Reviewer #1 (Remarks to the Author):

Overall, the work is very interesting, well written clearly reporting the research line carried out by the groups.

The work is well sustained by all the provided tables and figures.

Therefore, in my opinion it can be accepted in this form.

Reviewer #2 (Remarks to the Author):

This paper studied the transcriptional and functional role of immune cells in PBMC and BM samples of three conditions (HC, MNG and thyroid cancer). The paper is overall clearly written and data presented are generally supporting the conclusions. Some experiments were repeated with multiple stimulations for cytokine production and ROS responses. The results are of interest to the field. Methods used for analyses are appropriate.

Some comments may worth considerations:

The scRNAseq data were analyzed for cell populations and differential genes and results are appropriately presented. However, large variabilities in scRNAseq deconvoluted cell population frequencies are mostly observed. It is preferable to have AN orthogonal validation on the cell abundance analyses, e.g., using flow cytometric analysis.

All the findings are based on bioinformatic analyses but no mechanistic experiments for validation, for example, confirming downregulation of the oncogene AREG in myeloid/multipotent progenitor cells from TC and MNG patients compared to HC.

For the longitudinal analysis across time points (baseline, surgery, post time points), "two-way ANOVA" was used. But Repeated measures two-way ANOVA should be the correct method for the data in Graphad Prism as measurements were taken on each specimen repeatedly at different time point before and after treatment.

Varying sample size even for data of the same types of specimens, for example, Fig1 (HC n=8, MNG n=13, TC n=12) and Fig3 (HC n=7, MNG n=13, TC n=13), a brief explanation will be needed to understand the reason. Is this because of outlier removal based on the Grubb test? Will the results differ if the outlier is not removed? Some times, it is hard to tell whether it is an outlier or simply due to biological variability.

For ROS assay statistical analysis, "ROS production was calculated as area under the curve (AUC) of luminescence signal during a measurement of 1h. Data was corrected for age and sex." So, the AUC data was derived after correcting for age and sex and then the adjusted AUC was compared by 1-way ANOVA? It is not clear how the data was corrected for age and sex, using what model or what software?

1-sided tests were used in some places (e.g., Fig2J clearly stated that Wilcoxon 1-sided test was used). In general, 2-sided p which should be used to be more conservative.

GO term pathway analysis (e.g., Fig1B) , not sure what is GeneRatio2 and how to interpret it?

Please include in method on What was used to generate the pseudo time plot in Fig1?

Fig1C GO term analysis the image color indicating p values are not quite differentiable.

Page10 Line 171-173, redundant texts.

Reviewer #3 (Remarks to the Author):

Rabold et al. analyzed scRNA-seq data obtained from PBMC and BM cells of patients with thyroid cancer and multinodular goiter. The major findings include functional differences in classical monocytes of thyroid cancer patients summarized by immunological tolerance and over production of ROS. These functional changes were subsequently found along healthy controls-benign goiters-TCs. They also analyzed PBMC samples after surgery and I-131 therapy. After sorting CD14+C16-monocytes, cytokine release and ROS production tests were performed. The results showed that ROS production decreased after I-131 treatment, while cytokines production was not changed

after operation or I-131 treatment.

This is an interesting study examining how the systemic innate immune system is related to tumors through scRNA-seq from human samples. In addition, BM sampling was also included, it supported functional changes in BM-derived myeloid cells. Furthermore, it is very interesting to try to validate the hypothesis through a functional study after treatment, surgery and I-131 treatment. Nevertheless, there are some limitations and critical points that should be revised.

- The authors conclude that changes in myeloid cells occur prior to tumor infiltration. However, the results do not support this. Perhaps it was taken into account that a moderate level of functional changes analyzed from scRNA-seq was also observed in the benign multinodular goiter. However, multinodular goiter is not a precancer lesion of thyroid cancer, nor does thyroid cancer necessarily accompany MNG.

- To say that ROS function is "restored" after I-131 treatment is difficult to interpret as 'restoration'. Probably, I-131 treatment was performed for the purpose of ablation in most patients (Rather than treatment of thyroid cancer lesions except TC patients with metastasis). That is, the tumors has been grossly removed by surgery, and the purpose of I-131 treatment would be the removal of remnant thyroid tissue. At this time, the change in myeloid cells may be a direct BM effect of I-131 rather than tumor-related changes, or an inflammatory response related to the removal of normal remnant thyroid tissue.

- Regarding the sc-RNAseq data, It seems that data integration (i.e. batch effects removal) was not included (incorporated in Seurat). I wonder why you didn't try that.

- Log FC was not considered in DEG, so I am curious as to why it was implemented that way. When only the p-value is considered, a few differently expressed genes show very little effect size (i.e. logFC) despite the significant p-value due to the large number of cells, so, they could be 'biologically or clinically' insignificant..

- Perhaps several technical factors are involved, but the quality of the BM derived cells from which about 500 cells were produced per sample has an issue of reliability in interpretation. In particular, considering the cells for each sample of about 500, it is questionable whether the comparison of cell proportion can be performed among groups due to the reason that there are many cell type losses. Similarly, the difference in pseudo-bulk data also depends on the quality including the proportion of lost cells, and may not reflect the difference between groups.

- Minor points: The abbreviation of MNG is not mentioned in the text.

Reviewer #4 (Remarks to the Author):

This work innovatively explored the proportion and function changes of immune cells (or progenitor cells) from peripheral blood and bone marrow between thyroid cancer patients, multinodular goiter patients, and healthy controls, and further demonstrated the influences of treatment on these changes.

The design of this project is interesting and original, which may provide some valuable information to immunotherapy for thyroid carcinoma. However, some parts could be improved and I will list it below, and overall the scientific value is limited.

Major points:

1. In Immune cells are transcriptionally different between groups part, author discovered the different enrichment of pathways between TC, MNG, and HC in myeloid and lymph cells. Author could discuss more about these changes and give possible mechanisms according to the biological and clinical features of disease itself.

2. In the last paragraph of The impact of TC on bone marrow immune cells part (line 135-148), it compared the transcriptome signature between TC and HC in different cell types and concluded a closeness between CD14+ cells from BM and PBMC of TC in transcriptome level. I am confused

about the biological significance of the phenomenon. Besides, whether the closeness was a result of methodology, which I mean the method you use to get the signature. I am curious that If you conduct a new signature by comparing HC to TC, you may get a closeness between BM and PBMC of HC.

3. In Diminished amplification of cytokine production by cancer cells in myeloid cells of TC patients part, author used a trans-well system to verify a reduction of cytokine production in monocyte. However, as far as I am concerned, there is a flaw about the experiment design. What is your purpose to co-culture all monocytes from TC, MNG and HC with TPC-1 cells, and what clinical significance can we get from the reduction of cytokine?

4. In The impact of treatment on functional phenotype of peripheral blood monocytes part, it showed a reduction of ROS production in TC monocytes after RAI treatment, I doubt that the reduction may be a result of dose-dependent effect because TC patients received higher dose than MNG patients. Could this be tested through in vitro experiment?

Minor points:

1. I noticed that titles of results were showed in different fonts. Standardized and stratified fonts are strongly recommended.
2. In line 224-225, it should list references to support the statement.
3. in line 291-292, this sentence is a bit confused, please revise it to express more specific.

Reviewer #5 (Remarks to the Author):

Note: I was asked to review only the scRNA-seq analysis part of the paper

In this study the goal is to study the properties of innate immune cells in thyroid carcinoma (TC), in particular the myeloid cell lineage. To this end the authors analyse PBMCs and bone marrow samples from healthy controls and TC patients and (for reasons that are not explained) MNG samples (which I had a hard time to find out what that actually is).

They claim (in the abstract) from this scRNA-seq analysis that circulating monocytes from TC patients show upregulation of antigen presentation and in the bone marrow compartment and that TC patients show a different composition of progenitor cells.

On the positive side, I like the experimental design with $N > 1$ patients pooled and then deconvoluted using SNPs. However, the statistical basis how they reach the conclusions is nevertheless unclear or at least not properly reported. 1) it is not clear from the methods nor the results section whether the differentially expressed genes are identified with or without considering the individual sample. As the independent replicates are the samples (and not the cells) this is crucial. I think the FindMarker function of Seurat does not take this into account. I know that it is common practice, but I also know that this is very problematic as no claim on a patient population can be made when assuming cells are independent measures.

How they reach the claim that TC patients show a different composition of progenitor cells is even less clear to me. In the Figure legend for 2D they write: "Boxplots of cell proportions within the CD34+ progenitor cell populations derived from the bone marrow. No significant differences were found." They also report some trends in the text, but again it is not clear what independent replicates are assumed and its also not clear how they deal with the multiple testing issue as there are several cell types and three patient groups.

Independently, of the claims in the abstract that are not supported in my view, the bone marrow analysis is rather convoluted (especially from line 135 onwards). It is also entirely unclear to me how the described analyses relate to the initial question of monocyte properties in TC patients.

I am aware that many scRNA-seq analyses suffer from these problems. I also think that this is a nice dataset for TC patients. However, I also think that claims in the abstract should be statistically valid..

Reviewer 1:

Q1: Overall, the work is very interesting, well written clearly reporting the research line carried out by the groups.

The work is well sustained by all the provided tables and figures.

Therefore, in my opinion it can be accepted in this form.

R1: *We thank the reviewer for the compliments and taking the time to critically evaluate the manuscript.*

Reviewer 2:

Q1: This paper studied the transcriptional and functional role of immune cells in PBMC and BM samples of three conditions (HC, MNG and thyroid cancer). The paper is overall clearly written and data presented are generally supporting the conclusions. Some experiments were repeated with multiple stimulations for cytokine production and ROS responses. The results are of interest to the field. Methods used for analyses are appropriate.

R1: *We thank the reviewer for the positive assessment of the manuscript.*

Q2: Some comments may worth considerations:

The scRNAseq data were analyzed for cell populations and differential genes and results are appropriately presented. However, large variabilities in scRNAseq deconvoluted cell population frequencies are mostly observed. It is preferable to have AN orthogonal validation on the cell abundance analyses, e.g., using flow cytometric analysis.

R2: *We agree with the reviewer that additional validations of immune cell populations are very important and that such results should not be based solely on scRNAseq data.*

Firstly, we would like to kindly note that no deconvolution methods were used in determining the cell population frequencies in the single-cell data. Cells were clustered in an unsupervised manner, followed by automated cluster annotation (SingleR, Aran et al (2019), Nature Immunology) and manual annotation using canonical markers. The population frequencies are a direct result of the cell type annotation that was performed, and not deconvoluted.

Following the reviewer's suggestion, we have used cytometric data to validate cell population frequencies from the single cell data (Fig. 1 shown below). The single-cell approach and orthogonal validation have yielded consistent results. In the cytometric data, we show that the ratio of common myeloid progenitors (CMP) to multi-lymphoid progenitors (MLP) shows a decreasing trend in TC and MNG patients compared to HC. Furthermore, the number of common lymphoid progenitors (CLP) are significantly higher in MNG compared to HC, and show an increased trend in TC compared to HC. Finally, the abundance of megakaryocyte-erythroid progenitors (MEP) shows an increasing trend in HC compared to both TC and MNG.

Together, these results support our previously reported results, and suggest a shift from myelopoiesis towards lymphopoiesis in TC patients. We have added these results in Supplementary Figure 2 in the revised manuscript, and the following sentence in the revised manuscript:

"These results have been further validated by flow cytometric analysis (Supplementary Figure 2)" (Page 7 in the revised manuscript, Lines 119-120)

Figure 1: Flow cytometry quantification of bone-marrow derived cell populations. CMP: common myeloid progenitors. MLP: multi-lymphoid progenitors. CLP: common lymphoid progenitors. MEP: megakaryocyte-erythroid progenitors.

Q3: All the findings are based on bioinformatic analyses but no mechanistic experiments for validation, for example, confirming downregulation of the oncogene AREG in myeloid/multipotent progenitor cells from TC and MNG patients compared to HC.

R3: *We cannot agree more with the reviewer, and biological validation is an important component of any such study. Regarding biological validations in bone marrow cells themselves, we were unfortunately limited by the availability of patient material: as one can understand, it is very difficult to obtain such samples from patients, and all material has been used for purification and sequencing protocols. We also aimed to perform functional validations of the findings, and the experimental studies using primary circulating monocytes in TC patients are a clear functional validation of the immunological disturbances suggested by the transcriptome studies: indeed, monocytes in the circulation are directly descendent for the myeloid cell progenitors (they were in the bone marrow 1-2 days before being collected from the blood). We have now underlined this aspect more clearly in the revised manuscript (Page 14, lines 288 - 301).*

Q4: For the longitudinal analysis across time points (baseline, surgery, post time points), “two-way ANOVA” was used. But Repeated measures two-way ANOVA should be the correct method for the data in Graphad Prism as measurements were taken on each specimen repeatedly at different time point before and after treatment.

R4: *We thank the reviewer for this suggestion, and we agree with the reviewer that repeated two-way ANOVA is the correct method for the analysis. However, the patients in the multinodular goiter group consisted of two groups that received either surgery as treatment or radioactive iodine treatment. Therefore, we lack data for some timepoints in a few individuals because of the differences in treatment regimens. Furthermore, some data were also missing due to lack of enough biological material for all of the investigations that we wanted to perform. Therefore, it was unfortunately not possible to perform a reliable ‘repeated measures two-way ANOVA test’ on the same number of samples from all time points. For this reason, we decided to present the data of all measurements with two-way ANOVA. Regarding the ROS production, our dataset of repeated measures was unfortunately too small to perform a repeated measures two-way ANOVA, as this read-out was the lowest rank on our priority list and for many patients we lacked sufficient material. These are the objective constraints for using two-way ANOVA.*

To fully answer the reviewer's question, we have repeated our analysis using the repeated measures two-way ANOVA separately for surgery and radioactive treatments for the cell counts and cytokine production data. The repeated measures two-way ANOVA shows similar trends (Fig. 2 shown below). Specifically, for the cell count data, we observed a significant reduction in lymphocyte cell counts following I^{131} treatment, as reported in the manuscript. For the cytokine data upon LPS stimulation, we observed that neither treatment nor surgery had a large impact on the cytokine production upon stimulation.

These results have been included in Supplementary Figure 4, and we have added these results in the revised manuscript at Page 11, lines 226-228 as follows:

"Repeated two-way ANOVA test further confirmed similar patterns of the effect of treatment on immune cells numbers and function (Supplementary figure 4)"

Figure 2: **A)** Cell count data analyzed separately for surgery-treated (left- column) and I131-treated samples (right column). Repeated measures two-way ANOVA after exclusion of individuals with missing values. **B)** Analysis of cytokine data upon LPS stimulation using repeated measures two-way ANOVA after exclusion of samples with missing data.

Q5: Varying sample size even for data of the same types of specimens, for example, Fig1 (HC n=8, MNG n=13, TC n=12) and Fig3 (HC n=7, MNG n=13, TC n=13), a brief explanation will be needed to understand the reason. Is this because of outlier removal based on the Grubb test? Will the results differ if the outlier is not removed? Sometimes, it is hard to tell whether it is an outlier or simply due to biological variability.

R5: We thank the reviewer for this comment. The differences in number of specimens are not due to outlier removal. Unfortunately, we were limited by the logistic complexity of such a clinical study. The lack of sufficient biological material or number of cells for some measurements in some individuals have led to differences in sample sizes between read-outs of the same specimen type. We have tried to transparently describe the number of samples included in each analysis throughout the manuscript. Grubbs' test was only used for the proximity extension assay (PEA) of the circulating inflammatory markers, as this was a bigger dataset containing 75 inflammatory markers. We have clarified these issues in the revised manuscript as follows:

“For some of the immunological assays performed, not enough biological material was available for some of the patients, and subsequently the final number of tests may differ for the various assessments. The precise sample size is disclosed for each type of assay. No sample or outlier removal was performed for any of the immunological functional assays”. (Page 5, lines 75-78).

Q6: For ROS assay statistical analysis, “ROS production was calculated as area under the curve (AUC) of luminescence signal during a measurement of 1h. Data was corrected for age and sex.” So, the AUC data was derived after correcting for age and sex and then the adjusted AUC was compared by 1-way ANOVA? It is not clear how the data was corrected for age and sex, using what model or what software?

R6: *We apologize if the text was unclear. We have clarified these issues in the revised manuscript as follows.*

“ROS production was calculated as area under the curve during a measurement of 1h. Data was corrected for age and sex by fitting a linear model that included age and sex as covariates. The residuals from this model were extracted and used for further statistical analyses.” (Page 24, lines 535-537)

Q7: 1-sided tests were used in some places (e.g., Fig2J clearly stated that Wilcoxon 1-sided test was used). In general, 2-sided p which should be used to be more conservative.

R7: *We fully agree with the reviewer that two-sided tests should be used to be more conservative. Following the reviewer’s suggestions, we have repeated the analysis using a two-sided Wilcoxon test. All significant results reported in the initial manuscript remain significant using a two-sided test. We have adjusted these methods in the revised manuscript:*

“Enrichment in TC compared to HC was tested using a two-sided Wilcoxon test per cell type individually. P-values were adjusted (BH) to account for the multiple testing problem over all cell types together.” (Page 8, lines 145-147).

Q8: GO term pathway analysis (e.g., Fig1B), not sure what is GeneRatio2 and how to interpret it?

R8: *We apologize for the textual mistake. “GeneRatio2” was meant to be “GeneRatio”, representing the number of input genes annotated to a specific GO term, divided by the total number of genes annotated to this term. We have corrected this mistake and clarified the term “GeneRatio” in the revised manuscript:*

“The GeneRatio represents the number of input genes annotated a specific GO term, divided by the total number of genes annotated to this term” (Page 23, lines 507-508).

Q9: Please include in method on What was used to generate the pseudo time plot in Fig1?

R9: *The method employed to generate the pseudo-time figure was Monocle3, v0.2.3. This information shown below was included in the manuscript at page 23, line 514.*

“Cell type transition using each cell’s pseudotime was estimated using Monocle3 v0.2.3”

Q10: Fig1C GO term analysis the image color indicating p values are not quite differentiable.

R10: *We thank the reviewer for the comment. We have corrected the figure by adjusting the color scale as requested by the reviewer.*

Q11: Page10 Line 171-173, redundant texts.

R11: *We thank the reviewer for the comment and we corrected the text as follows:*

“In contrast, IL-1 β responses significantly decreased in TC patients and HC, but remained unchanged in MNG patients. (Page 9 lines 180-181). “

Reviewer #3 (Remarks to the Author):

Q1: Rabold et al. analyzed scRNAseq data obtained from PBMC and BM cells of patients with thyroid cancer and multinodular goiter. The major findings include functional differences in classical monocytes of thyroid cancer patients summarized by immunological tolerance and over production of ROS. These functional changes were subsequently found along healthy controls-benign goiters-TCs. They also analyzed PBMC samples after surgery and I-131 therapy. After sorting CD14+C16-monocytes, cytokine release and ROS production tests were performed. The results showed that ROS production decreased after I-131 treatment, while cytokines production was not changed after operation or I-131 treatment.

This is an interesting study examining how the systemic innate immune system is related to tumors through scRNAseq from human samples. In addition, BM sampling was also included, it supported functional changes in BM-derived myeloid cells. Furthermore, it is very interesting to try to validate the hypothesis through a functional study after treatment, surgery and I-131 treatment. Nevertheless, there are some limitations and critical points that should be revised.

R1: We thank the reviewer for thoroughly reviewing our manuscript, as well as for the positive assessment and for his/her valuable suggestions.

Q2: The authors conclude that changes in myeloid cells occur prior to tumor infiltration. However, the results do not support this. Perhaps it was taken into account that a moderate level of functional changes analyzed from scRNAseq was also observed in the benign multinodular goiter. However, multinodular goiter is not a precancer lesion of thyroid cancer, nor does thyroid cancer necessarily accompany MNG.

R2: We thank the reviewer for raising this relevant aspect. The hypothesis at the basis of our study is that in the process of carcinogenesis there is a continuous cross-talk between tumor cells on the one hand and immune cells on the other hand. While in the tumor microenvironment local factors such as changes in the metabolic milieu (among others) have been shown to mediate such cross-talk, we hypothesized that soluble tumor-derived factors also influence the developmental trajectory of bone marrow immune cell progenitors: practically, the immunosuppressive phenotype of myeloid cells in cancer is determined at a much earlier stage than currently thought. This hypothesis is supported by the current data in TC: what we meant by 'tumor infiltration' is that no tumor cells are present in the bone marrow of the patients, and TC in these patients is not metastasized, yet important changes are present in immune cell progenitors in the bone marrow.

An additional question is whether such changes already occur before neoplastic transformation of multinodular goiter (MNG). The reviewer is correct, MNG does not represent precancerous lesions, and we have also mentioned this in the Discussion of the manuscript (Page 13, lines 260-265). We do find however changes also in cells isolated from goiter patients, which suggests that such changes in immune cells can also be triggered by other neoplastic lesions as well. This also underscores the importance of including the patients with MNG as a comparator instead of only healthy volunteers. Nonetheless, we do suspect that these changes could be even more pronounced in patients with extensive metastatic disease, who could likely benefit more from a targeted immunological intervention. We currently explore these hypotheses in ongoing studies.

Q3: To say that ROS function is "restored" after I-131 treatment is difficult to interpret as 'restoration'. Probably, I-131 treatment was performed for the purpose of ablation in most patients (Rather than

treatment of thyroid cancer lesions except TC patients with metastasis). That is, the tumors have been grossly removed by surgery, and the purpose of I-131 treatment would be the removal of remnant thyroid tissue. At this time, the change in myeloid cells may be a direct BM effect of I-131 rather than tumor-related changes, or an inflammatory response related to the removal of normal remnant thyroid tissue.

R3: We thank the reviewer for this correct observation that the effect seen on ROS production could be due to either iodine treatment itself, or the elimination of the tumor load (and thus the tumor-derived mediators influencing bone marrow progenitors). In order to discern between these two possibilities, we performed an in-vitro experiment in which we exposed human primary monocytes to different concentrations of iodine in the context of cytokine and ROS stimulation assays as performed in the patients. We have added this description in the revised version as follows:

“To exclude a direct effect of iodine on cytokine and ROS production capacity, we stimulated monocytes with LPS (24h for cytokine production) or zymosan (1h for ROS production) in the presence of different concentrations of iodine (1, 10 and 50 nM). Iodine did not influence production of cytokines or ROS when 1 and 10 nM concentrations were used, with only a moderate effect on TNF (Supplementary Figure 5). Higher concentrations of iodine (50nM) were toxic to the cells (not shown). These data argue that the changes of cytokine and ROS production after treatment of patients are most likely due to the removal of the tumor (and thus tumor-associated mediators), rather than the effects of iodine treatment itself.” (Page 11, lines 228-235)

We did not observe a significant effect of iodine on cytokines and ROS production, with a small effect on TNF release. While an in-vitro assay has its limitations, this argues that the changes of cytokine and ROS production after treatment of patients are most likely due to the removal of the tumor (and thus tumor-associated mediators), rather than the effects of iodine treatment itself. We added these data on Page 15, but also toned down the claim and rephrased the statement accordingly: “Interestingly, after treatment with ¹³¹I, ROS production of circulating classical monocytes in TC patients decreased to the levels similar to those found in the healthy volunteers (Page 15, lines 239331)”.

Q4: Regarding the scRNAseq data, it seems that data integration (i.e. batch effects removal) was not included (incorporated in Seurat). I wonder why you didn't try that.

R4: We thank the reviewer for this feedback. During the initial data exploration, we observed only minimal batch effects derived from the different sequencing pools. All minor and major cell populations were retrieved in every pool (Fig. 3, shown below). We did notice that pool 10 seems to be slightly different compared to other pools, therefore we carefully checked cells derived from this pool. This pool is comparable with all the other pools in terms of the number of genes and number of reads detected per cell. Furthermore, since we fully randomized the different conditions (TC, MNG and HC) and tissues (PBMC, BMMC) over the pools, we believe that any pool-derived batch effect in downstream analyses is minimal.

To fully answer the reviewer's question, we have applied Harmony [Korsunsky et al (2019), Nat. Methods] to remove the potential pool batch effect and compared this to our previous results. We annotated the cell types anew based on the batch-corrected data (Fig. 4A). Cell proportions were extremely consistent whether we employed batch effect correction or not (Fig. 4B). Finally, we assessed the overlap between the cell type annotations before and after integration and showed that the cell type annotation per cell is almost identical before and after integration (Fig. 4C). The proportion of cells annotated to each celltype was consistent. Together, these coloring the cells by cell type annotation, it is clear that the same cells cluster together based on their transcriptional similarities (Fig. 4, shown below). Clustering and subsequent annotation of the cells is not dependent on harmonization of the

data, and thus the downstream results are also not affected. These results further confirm that the batch effect in this dataset is minimal.

We have clarified this in the revised manuscript at Page 20, lines 432-433: “We did not observe any significant batch effects between the different sequencing pools”.

Fig. 3: Distribution of cells per pool over the entire dataset. Per pool, cells derived from a specific pool are highlighted in red.

C

Fig. 4: A) UMAP dimensionality reduction without (left) and with (right) Harmony integration on the sequencing pool effect. Cells are colored by their cell type annotation. B) Barplots showing the distributions of cell populations across samples with (left) and without (right) Harmony integration. Each bar represents a sample, bars are colored by celltype. C) Overlaps between the celltype annotation before and after Harmony integration.

Q5: Log FC was not considered in DEG, so I am curious as to why it was implemented that way. When only the p-value is considered, a few differently expressed genes show very little effect size (i.e. logFC) despite the significant p-value due to the large number of cells, so, they could be 'biologically or clinically' insignificant.

R5: *The reviewer is correct, and following this suggestion we repeated the differential expression analyses with a logFC cutoff of 0.25. The results obtained from these analyses are convincingly similar to the earlier results, showing that the differences we report are based on true biological observations. To illustrate this, we show below GO term enrichment results as reported in the original paper (Fig 5 shown below) and the corresponding enrichment results when applying a logFC cutoff on the differentially expressed genes (Fig. 6 shown below). Both figures show enrichment in biologically similar pathways, specifically ATP metabolism-related processes, T cell-related processes and response to interferon gamma.*

Fig. 5: GO term enrichment of all significantly differentially expressed genes in selected cell types (no logFC cutoff applied).

Fig. 6: GO term enrichment of significantly differentially expressed genes in selected cell types when employing a logFC cutoff of 0.25.

We have accordingly updated enrichment figures (Fig. 1D) as well as single-cell RNA sequencing volcano plots (Fig. 2H). Furthermore, we have clarified the logFC threshold that was used in the Methods section at Page 23, lines 503-505: “We specifically used the MAST⁴⁸ test while correcting for sample age and sex. P values were adjusted using Bonferroni correction, and a log fold change threshold of 0.25 was applied”.

Q6: Perhaps several technical factors are involved, but the quality of the BM derived cells from which about 500 cells were produced per sample has an issue of reliability in interpretation. In particular,

considering the cells for each sample of about 500, it is questionable whether the comparison of cell proportion can be performed among groups due to the reason that there are many cell type losses. Similarly, the difference in pseudo-bulk data also depends on the quality including the proportion of lost cells, and may not reflect the difference between groups.

R6: The reviewer is indeed correct, in some patients we were restricted by the quantity of the clinical samples received. While we have extensive experience with single-cell RNA sequencing procedures and analysis, we cannot fully exclude certain cell type losses during procedures, and we added this point in the 'limitations' part of the Discussion at Page 16, lines 348-351:

"Quality and reliability of our pseudobulk analyses rely on the underlying single-cell results. While we have not observed batch effects of quality artefacts, pseudobulk analyses could be influenced by technical factors such as specific cell type losses."

However, per total the data were reproducible and of very good quality, which argues for the robustness of our conclusions. In addition, we now also provide cell count analysis using classical flow cytometry methods that complement the transcriptome-based assessments (please see our answer to reviewer #2, Q2, and Supplementary Figure 2)

Q7: Minor points: The abbreviation of MNG is not mentioned in the text.

R7: We apologize for this omission. We have explained the MNG abbreviation in the revised manuscript (Page 4, Line 63).

Reviewer #4 (Remarks to the Author):

Q1: This work innovatively explored the proportion and function changes of immune cells (or progenitor cells) from peripheral blood and bone marrow between thyroid cancer patients, multinodular goiter patients, and healthy controls, and further demonstrated the influences of treatment on these changes. The design of this project is interesting and original, which may provide some valuable information to immunotherapy for thyroid carcinoma. However, some parts could be improved and I will list it below, and overall the scientific value is limited.

R1: *We thank the reviewer for carefully reviewing our manuscript and for the positive assessment of the originality and relevance of our study. We would like to point out that this is a unique valuable dataset that not only provides comparisons between the immune cells function in patients with malignant tumors and healthy volunteers, but also introduces the group of patients with benign thyroid tumors as an additional comparator. The latter is particularly relevant as those patients require similar treatments as cancer patients do. Though clearly acknowledging the limitations of our study in the Discussion, we still believe that the robust single cell analysis combined with the functional studies in this patient populations confers the study significant scientific value and an important starting point of future studies to extend the investigations to patient categories that are expected to benefit the most from a targeted intervention on the function of the immune cells. We also believe that these findings can help shaping future targeted immunological interventions in cancer patients.*

Major points:

Q2. In Immune cells are transcriptionally different between groups part, author discovered the different enrichment of pathways between TC, MNG, and HC in myeloid and lymph cells. Author could discuss more about these changes and give possible mechanisms according to the biological and clinical features of disease itself.

R2: *We thank the reviewer for this recommendation. As suggested by the reviewer, in the Discussion of the revised manuscript, we have elaborated more on the potential mechanisms and implications of the findings within the biological and clinical context of the disease as follows:*

*“A reduced cytokine production capacity of monocytes from TC patients may possibly be induced by circulating tumor-derived mediators that change their function or that of their precursors in the bone marrow: this hypothesis needs to be investigated in future studies. Moreover, upon exposure to factors secreted by a TC cell line in a trans-well model *ex-vivo*, classical monocytes from TC patients show a strongly impaired secretion of IL-6 compared to cells isolated from MNG patients, confirming the presence of a suppressed cytokine production capacity in response to tumor-derived factors. This might indicate a decreased ability of monocytes from TC patients to respond efficiently after the encounter with tumor cells.” (Page 14 lines 303-310).*

Q3. In the last paragraph of the impact of TC on bone marrow immune cells part (line 135-148), it compared the transcriptome signature between TC and HC in different cell types and concluded a closeness between CD14+ cells from BM and PBMC of TC in transcriptome level. I am confused about the biological significance of the phenomenon. Besides, whether the closeness was a result of methodology, which I mean the method you use to get the signature. I am curious that If you conduct a new signature by comparing HC to TC, you may get a closeness between BM and PBMC of HC.

R3: *We thank the reviewer for raising this very relevant point, and we apologize that we did not describe it more clearly in the manuscript. The shared signature between the transcriptional profiles of CD14+ cells in bone marrow and circulation is indeed very relevant biologically: this basically demonstrates that CD14+ cells already have a changed transcriptional program in the bone marrow, before entering the circulation. Subsequently, this becomes also important for future therapeutic approaches against immunosuppressive CD14+ cells: such immunotherapies to be effective need to target also the bone marrow. We have better underlined these aspects in the revised manuscript (Page 14, line 288 - 301).*

Furthermore, the reviewer makes a good suggestion to compare CD14+ monocytes from the bone marrow and circulation in healthy individuals. In our analysis presented in the manuscript, we have compared the similarities between transcriptomes from the bone marrow and circulation, using all genes significantly differentially expressed between TC and HC per compartment. The method we have used (AUCell, Aibar et al (2019), Nat. Methods) does not consider the direction of a specific gene, meaning the signature TC vs HC is identical to HC vs TC. As the signatures are identical, the enrichment results would also be precisely identical, showing higher enrichment in TC compared to HC, as reported in the initial manuscript.

In addition, we suggest the following interpretation of the results: when considering genes that are significantly different between TC and HC, we observe that transcriptomes from CD14+ monocytes are significantly more similar between the bone marrow and circulations in TC patients compared to HC. As outlined above, this suggests that the CD14+ monocytes have already changed their transcriptional signature in the bone marrow before entering the circulation. We have added more interpretation on these findings in the revised manuscript at Page 14, lines 294-298: “The shared signature between CD14+ monocytes in the bone marrow and the circulation suggest that CD14+ monocytes in the bone marrow have already changed their transcriptional program before entering the circulation. Future therapeutic approaches against immunosuppressive CD14+ monocytes thus need to also target the bone marrow-resident cells in order to reach full efficacy.”

Q4: In Diminished amplification of cytokine production by cancer cells in myeloid cells of TC patients part, author used a trans-well system to verify a reduction of cytokine production in monocyte. However, as far as I am concerned, there is a flaw about the experiment design. What is your purpose to co-culture all monocytes from TC, MNG and HC with TPC-1 cells, and what clinical significance can we get from the reduction of cytokine?

R4: *We regret the mistake in the title of the paragraph: we did **not** mean the cytokine production of the cancer cells. We have corrected the title of the paragraph (Page 9, line 174) and we have provided the additional clarification in the revised manuscript. In addition, the purpose of the co-culture experiment is to assess the cytokine production capacity by the immune cells, as a readout for the function of the circulating monocytes from patients with TC, MNG and HC, when these cells have been exposed to thyroid cancer cell lines. This experiment design provides a more reliable assessment of the cytokine production capacity of these cells than the measurement of cytokine concentrations in circulation, which is dependent on many other factors and might not necessarily reflect the function of these cells in the context of cancer. The cytokine production of the monocytes from healthy volunteers is amplified after co-culture with cancer cell lines. The fact that we found a diminished amplification of cytokine production capacity upon ex-vivo exposure to thyroid cancer cell lines in patients with TC indicates a poorer response of immune cells from TC patients to soluble tumor-derived factors (an immunosuppressed phenotype). We explained these aspects more clearly in the revised manuscript (Page 14, lines 303-310).*

Q5: In The impact of treatment on functional phenotype of peripheral blood monocytes part, it showed a reduction of ROS production in TC monocytes after RAI treatment, I doubt that the reduction may be a result of dose-dependent effect because TC patients received higher dose than MNG patients. Could this be tested through in vitro experiment?

R5: *We thank the reviewer for this suggestion. We performed an in-vitro experiment in which we exposed human primary monocytes to iodine in the context of cytokine stimulation assays as performed in the patients. As described in the revised version of the manuscript on Page 11 and shown in Supplementary figure 5, we did not observe a significant effect of iodine on cytokines and ROS production, with a small effect on TNF release. While an in-vitro assay has its limitations, this argues that the changes of cytokine and ROS production after treatment of patients are most likely due to the removal of the tumor (and thus tumor-associated mediators), rather than the effects of iodine treatment itself. We added these data to the supplementary of the manuscript (supplementary figure 5) and we discussed this in the revised manuscript (Page 11, line 228-235 and Page 16 337-341).*

Minor points:

Q6. I noticed that titles of results were showed in different fonts. Standardized and stratified fonts are strongly recommended.

R6: *We have changed the fonts accordingly and updated figures in the revised*

manuscript. **Q7:** In line 224-225, it should list references to support the statement.

R7: *In the Discussion lines 244-245, the hypothesis of the study is presented. We now discussed the hypothesis in the revised manuscript in line with the new experiments (as described above). To the best*

of our knowledge, there are no studies yet published on this hypothesis, and our data provide the first new information on this.

Q8. in line 291-292, this sentence is a bit confused, please revise it to express more specific.

R8: We apologized for this confusion. Following the reviewer's comments, we have rephrased the sentence accordingly: "Interestingly, treatment with ¹³¹I reverted ROS production of circulating classical monocytes in TC patients towards the levels produced by monocytes of healthy controls." was replaced by: "Interestingly, after treatment with ¹³¹I, ROS production of circulating classical monocytes in TC patients decreased to the levels similar to those found in the healthy volunteers." (Page 15, Lines 329331)

Reviewer #5 (Remarks to the Author):

Note: I was asked to review only the scRNA-seq analysis part of the paper.

Q1: In this study the goal is to study the properties of innate immune cells in thyroid carcinoma (TC), in particular the myeloid cell lineage. To this end the authors analyse PBMCs and bone marrow samples from healthy controls and TC patients and (for reasons that are not explained) MNG samples (which I had a hard time to find out what that actually is).

R1: We thank the reviewer for taking the time to review our manuscript and for the appreciation of the originality of the study concept and design. We do apologize for omitting to name the abbreviation of multinodular goiter (MNG), which is unfortunate indeed. Multinodular goiters represent benign tumors of the thyroid gland. Because most of the studies use healthy volunteers as controls, we chose to include also this group of patients as a unique comparator, because some patients with multinodular goiter undergo some of the treatments (surgery or radioactive iodine) similar to those applied in patients with malignant TC. This gave us the opportunity to longitudinally examine the changes in immune cells phenotypes, along the different treatment trajectory. We now explained these aspects more clearly in the revised manuscript as follows:

"As comparators we used both healthy volunteers and patients with multinodular goiter (MNG), representing benign tumors of the thyroid gland that undergo treatments (surgery or radioactive iodine) similar to those applied in patients with malignant TC". (Page 4, lines 62-64).

Q0: They claim (in the abstract) from this scRNA-seq analysis that circulating monocytes from TC patients show upregulation of antigen presentation and in the bone marrow compartment and that TC patients show a different composition of progenitor cells.

On the positive side, I like the experimental design with N>1 patients pooled and then deconvoluted using SNPs. However, the statistical basis how they reach the conclusions is nevertheless unclear or at least not properly reported. 1) it is not clear from the methods nor the results section whether the differentially expressed genes are identified with or without considering the individual sample. As the independent replicates are the samples (and not the cells) this is crucial. I think the FindMarker function of Seurat does not take this into account. I know that it is common practice, but I also know that this is very problematic as no claim on a patient population can be made when assuming cells are independent measures.

R2: We thank the reviewer for considering our work very thoroughly. Firstly, we thank the reviewer for acknowledging our single-cell design. SNP-based deconvolution was indeed performed to assign cells to each individual, allowing us to obtain single-cell RNA-seq data in a cost-effective manner.

Secondly, we have identified differentially expressed genes without considering the individual sample, using well-established methodologies. Specifically:

We have used the R package Seurat for our single-cell analyses, which is one of the most commonly used tools in the field [Satija et al (2015) **Nature Biotechnology**, Butler et al (2018) **Nature Biotechnology**, Stuart et al (2019) **Cell**, Hao et al (2021) **Cell**, cumulative citations: >12.000]. Currently, Seurat provides methods to integrate covariates in a differential expression model, but only as fixed effects (e.g. age sex). Random effects such as the individual sample are not yet implemented and thus not available to the scientific community without manually adjusting the statistical procedures. This demonstrates that incorporating the individual sample in the differential expression model is not yet common practice, in line with the reviewer's comments.

We fully agree that the reviewer raises a valid point, and recognize that methods to properly incorporate the individual sample in single-cell statistics are needed. Currently, there is ample ongoing discussion in the scientific community concerning the pseudo-replication bias [Squair et al (2021), *Nature Communications*], and novel statistical methods are under development. However, we feel that developing a novel tool to incorporate the random effect of the individual sample is outside of the scope of the current work.

On the other hand, our sample sizes were unfortunately limited in this study due to logistical constraints (e.g. bone marrow from healthy individuals). From a statistical point of view, it may not be feasible for us to correct for the effect of the individual sample.

Finally, in order to fully answer the reviewer's question, we collapsed our single-cell data into pseudo-bulk data per patient and cell type individually. We would like to emphasize that the power in this analysis is severely limited and much lower compared to single-cell analyses corrected for the individual sample. We performed this analysis to show that the obtained differentially expressed genes in pseudo-bulk data are enriched for the same biological pathways as in the single-cell sequencing results reported in the original manuscript. We show below that the differentially expressed genes in this underpowered pseudo-bulk RNA seq analysis remain significantly enriched for the same pathways as reported in the original manuscript, related to T-cells and the regulation of interferon gamma. We propose to keep the description and methodology intact considering the identity of the results using both methods, but we would be happy to add this information in the manuscript if the editorial board think this would be useful.

Fig. 7: GO term enrichment of the significantly different genes in pseudo-bulk in CD8+ T cells.

Q3: How they reach the claim that TC patients show a different composition of progenitor cells is even less clear to me. In the Figure legend for 2D they write: “Boxplots of cell proportions within the CD34+ progenitor cell populations derived from the bone marrow. No significant differences were found.” They also report some trends in the text, but again it is not clear what independent replicates are assumed and it’s also not clear how they deal with the multiple testing issue as there are several cell types and three patient groups.

R3: *These are important aspects, and we thank the reviewer for pointing them out. Indeed, we find no significant differences, likely due to sample size limitations. As the reviewer noted, we describe trends for differences between groups in the manuscript. The trends we observe in the single-cell data are now validated using flow cytometry data (Please also see our answer to Reviewer #2, Question 2)*

We have analyzed the cell proportions in PBMCs, BMMCs and CD34+ progenitors by fitting a Dirichlet regression model. We have specifically chosen this model to account for the fact that cell proportions are not independent. The model was fitted on the cell proportions for each cell type, per comparison of two conditions separately (TC vs HC, TC vs MNG and MNG vs HC). Thus, this model quantifies the change in proportion per cell type between two conditions. To be on the conservative side, we accounted for multiple testing by adjusting the p-values using the stringent Bonferroni correct across all cell types and comparisons of conditions.

We have clarified this in the revised manuscript at Page 23, lines 510-513 : “We analyzed cell proportion differences by applying Dirichlet regression⁵⁰ using all cells, exclusively PBMC derived cells, bone marrow derived cells and progenitor cell types separately. The Dirichlet model was fitted on the cell proportions for each cell type, per comparison of two conditions separately (TC vs HC, TC vs MNG and MNG vs HC). P values were adjusted for multiple testing using Bonferroni correction across all cell types and comparisons of conditions.”

Q4: Independently, of the claims in the abstract that are not supported in my view, the bone marrow analysis is rather convoluted (especially from line 135 onwards). It is also entirely unclear to me how the described analyses relate to the initial question of monocyte properties in TC patients.

I am aware that many scRNAseq analyses suffer from these problems. I also think that this is a nice dataset for TC patients. However, I also think that claims in the abstract should be statistically valid.

R4: *We apologize if we did not describe the data clearly enough, and we cannot agree more that any claims need to be supported statistically. We have now described more clearly and in more detail the findings, also toning down some speculations. Regarding the relevance of transcriptome data in relation to the function of the monocytes, this is an excellent point raised by the reviewer. On Page 6, line 116 of the original manuscript we describe based on transcriptomic analysis that TC patients tend to have lower proportion of myeloid/multipotent progenitor cells compared to healthy controls, with a shift towards late B-cell progenitors. Indeed, trajectory analysis shows that in TC the multipotent progenitor cells tend to differentiate mainly towards B-cells, with a subsequent defect in myelopoiesis. Subsequently, these findings were validated by flow cytometry. These data, complemented with the immunosuppressed phenotype of circulating monocytes, argue for a continuum in the defective function of myeloid ontogeny and function in TC patients. We now discussed this more clearly in the revised manuscript both in the Abstract (Page 2, lines 11-13) and the Discussion (Page 14, lines 288301, Page 14, lines 303-310), but in a cautious tone necessary due to the relatively limited number of patients studied.*

REVIEWERS' COMMENTS

Reviewer #2 (Remarks to the Author):

The authors have addressed most of my previous comments.

For my previous comment Q4 on using the appropriate repeated measure model to analyze the longitudinal data, the authors implemented some analyses by excluding subjects with missing data, which is not preferred due to the small sample size. The conventional ANOVA may require balanced design (or sample size), but the more general linear mixed effects model, as implemented in SAS or R, or the repeated measure ANOVA in Graphad Prism, actually do allow subjects with missing values at some time point(s). See prism manual at https://www.graphpad.com/guides/prism/latest/statistics/stat_missing_values_and_two-way_ano.htm , and quoted below

"

Prism (starting with Prism 8) can also do the equivalent of repeated measures two-way ANOVA if values at some repeats are missing, so long as not too many points are missing and they are missing completely at random.

"

I suggest the authors can look into this issue.

Reviewer #3 (Remarks to the Author):

Most of the questions have been resolved. Most results became clearer.

- In vitro experiments (regarding ROS function) with iodine, however, are insufficient to support the results in human data. The effect of iodine may be due to beta radiation, not to the pharmacologic effect of iodine itself, which has not been tested in vitro. Also, as in the previous question, most patients do not have cancer cells left due to surgery and the purpose of radioactive iodine ablation of normal thyroid tissue is not associated with the effect of tumor burden. It is necessary to clarify the limitations of this part.

Reviewer #4 (Remarks to the Author):

The revised manuscript answered our question clearly, and gave experimental data to verify their results.

Thus, we thought it can be accepted.

Reviewer #5 (Remarks to the Author):

I think the authors did a good job in answering my concerns and incorporating more cautious wording and additional data (FACS). Hence, I think it is fine to go

Response to reviewer comments

Reviewer #2 (Remarks to the Author): The authors have addressed most of my previous comments.

For my previous comment Q4 on using the appropriate repeated measure model to analyze the longitudinal data, the authors implemented some analyses by excluding subjects with missing data, which is not preferred due to the small sample size. The conventional ANOVA may require balanced design (or sample size), but the more general linear mixed effects model, as implemented in SAS or R, or the repeated measure ANOVA in Graphad Prism, actually do allow subjects with missing values at some time point(s). See prism manual, and quoted below:

"

Prism (starting with Prism 8) can also do the equivalent of repeated measures two-way ANOVA if values at some repeats are missing, so long as not too many points are missing and they are missing completely at random.

"

I suggest the authors can look into this issue.

R: We thank the reviewer for this feedback and their time. As the reviewer suggested, we implemented a general linear mixed model in GraphPad Prism 8 to perform the analyses with missing data. Since the MNG patients either received surgery or treatment with radioactive iodine, a comparison between the 30 days post-surgery and the post-radioactive iodine treatments is not possible (all data points are missing). As this comparison is also not relevant to our research questions, we only compared the post-treatment time points to the baseline with the mixed model. Our conclusions from the analysis regarding cell counts and cytokine response did not change. Regarding the ROS production, the results of the new analysis slightly changed, probably due to the small number of samples and many missing values: as this read-out was the lowest rank on our priority list of the immunological assays, for many patients we lacked sufficient material. The adjusted analyses are depicted in the revised Fig. 4.

Reviewer #3 (Remarks to the Author): Most of the questions have been resolved. Most results became clearer.

- In vitro experiments (regarding ROS function) with iodine, however, are insufficient to support the results in human data. The effect of iodine may be due to beta radiation, not to the pharmacologic effect of iodine itself, which has not been tested in vitro. Also, as in the previous question, most patients do not have cancer cells left due to surgery and the purpose of radioactive iodine ablation of normal thyroid tissue is not associated with the effect of tumor burden. It is necessary to clarify the limitations of this part.

R: We are pleased to hear that the results have become clearer. To address the final question from this reviewer, we have clarified this limitation as requested by the reviewer:

Lines 476-409: "Nonetheless, we cannot completely exclude that the effect observed after ¹³¹I treatment may be due to the beta radiation, as most of the patients did not have large amounts of cancer cells remaining after the primary surgery."

Reviewer #4 (Remarks to the Author): The revised manuscript answered our question clearly, and gave experimental data to verify their results. Thus, we thought it can be accepted.

R: We thank the reviewer for their time and this positive feedback.

Reviewer #5 (Remarks to the Author): I think the authors did a good job in answering my concerns and incorporating more cautious wording and additional data (FACS). Hence, I think it is fine to go.

R: We thank the reviewer for this assessment.